# Unbalanced Optimal Transport through Non-negative Penalized Linear Regression

**Laetitia Chapel**\*
IRISA, Université Bretagne-Sud
Vannes, France
`laetitia.chapel@irisa.fr`

**Rémi Flamary**\*
CMAP, Ecole Polytechnique
Palaiseau, France
`remi.flamary@polytechnique.edu`

**Haoran Wu**
LITIS & IRISA
Rouen & Vannes, France
`haoran.wu@univ-ubs.fr`

**Cédric Févotte**
IRIT, Université de Toulouse, CNRS
Toulouse, France
`cedric.fevotte@irit.fr`

**Gilles Gasso**
LITIS, INSA Rouen Normandie
Rouen, France
`gilles.gasso@insa-rouen.fr`

## Abstract

This paper addresses the problem of Unbalanced Optimal Transport (UOT) in which the marginal conditions are relaxed (using weighted penalties in lieu of equality) and no additional regularization is enforced on the OT plan. In this context, we show that the corresponding optimization problem can be reformulated as a non-negative penalized linear regression problem. This reformulation allows us to propose novel algorithms inspired from inverse problems and nonnegative matrix factorization. In particular, we consider majorization-minimization which leads in our setting to efficient multiplicative updates for a variety of penalties. Furthermore, we derive for the first time an efficient algorithm to compute the regularization path of UOT with quadratic penalties. The proposed algorithm provides a continuity of piece-wise linear OT plans converging to the solution of balanced OT (corresponding to infinite penalty weights). We perform several numerical experiments on simulated and real data illustrating the new algorithms, and provide a detailed discussion about more sophisticated optimization tools that can further be used to solve OT problems thanks to our reformulation.

## 1   Introduction

Optimal Transport (OT) theory provides powerful tools for comparing probability distributions and has been successfully employed in a wide range of machine learning applications such as supervised learning (Frogner et al., 2015), clustering (Ho et al., 2017), generative modelling (Arjovsky et al., 2017), domain adaptation (Courty et al., 2017), learning of structured data (Maretic et al., 2019; Vayer et al., 2019) or natural language processing (Kusner et al., 2015), among many others. One reason for those recent successes is the introduction of entropy-regularized OT that can be solved with the efficient Sinkhorn-Knopp matrix scaling algorithm (Cuturi, 2013). However, the classical OT problem seeks the optimal cost to transport *all* the mass from a source distribution to a target one

---

\*First two authors have equal contribution

35th Conference on Neural Information Processing Systems (NeurIPS 2021).

(Villani, 2009), greatly limiting its use in scenarii where the measures have different masses or when they contain noisy observations or outliers.

Unbalanced Optimal Transport (UOT) (Benamou, 2003) has been introduced to tackle this shortcoming, allowing some mass variation in the transportation problem. It is expressed as a relaxation of the Kantorovich formulation (Kantorovich, 1942) by penalizing the divergence between the marginals of the transportation plan and the given distributions. Several divergences can be considered, such as the Kullback-Leiber (KL) divergence (Frogner et al., 2015; Liero et al., 2018), the $\ell_1$ norm corresponding to the partial optimal transport problem (Caffarelli and McCann, 2010; Figalli, 2010), or the squared $\ell_2$ norm (Benamou, 2003). Regarding numerical solutions, Chizat et al. (2018) considered an entropic-regularized version of UOT leading to a class of scaling algorithms in the vein of the Sinkhorn-Knopp approach (Sinkhorn and Knopp, 1967). The introduction of this entropic regularization improves the scalability of OT, but involves a spreading of the mass and a loss of sparsity in the OT plan. When a sparse transport plan is sought, the convergence is slowed down, necessitating the use of acceleration strategies (Thibault et al., 2021). Regarding UOT with the (squared) $\ell_2$ norm, Blondel et al. (2018) showed that the resulting OT plan is sparse and proposed to use an efficient L-BFGS-B algorithm (Byrd et al., 1995) to address this case. Note that the L-BFGS-B method can be used to solve UOT with differentiable divergences even without the entropic-regularization on the OT plan that induces the Sinkhorn-like iterations. Also note that, as for balanced OT, UOT can be solved more efficiently when the data has a specific structure, such as unidimensional distributions (Bonneel and Coeurjolly, 2019) or distributions supported on trees (Sato et al., 2020). Finally, recent work investigated UOT between Gaussians and provided closed form solutions for the regularized Janati et al. (2020) and unregularized (Janati, 2021, Eq. 2.72) versions of UOT associated with a KL divergence.

**Contributions.** In this paper, we show after some preliminaries that UOT can be recast as a convex penalized linear regression problem with non-negativity constraints (Section 2.2). The main interest of this reformulation resides in the fact that non-negative linear regression has been extensively studied in inverse problems and machine learning, offering a large panel of tools for devising new numerical algorithms. Our reformulation involves a design/dictionary matrix that is structured and sparse. Leveraging this structure, we propose two new families of algorithms for solving the exact (i.e., without regularization of the plan) UOT problem in Section 3.

We first derive in Section 3.1 a new Majorization-Minimization (MM) algorithm for solving UOT with Bregman divergences, and more specifically KL and $\ell_2$-penalized UOT. The MM approach results in multiplicative updates that have appealing features: i) they are easy to implement, ii) have low complexity per iteration and can be instantiated on GPU, iii) ensure monotonicity of the objective function and inherit existing convergence results. Our methodology is inspired by well-known algorithms in image restoration (Richardson, 1972; De Pierro, 1993) and non-negative matrix factorization (NMF) (Lee and Seung, 2001; Dhillon and Sra, 2005; Févotte and Idier, 2011). Interestingly, the resulting multiplicative updates bear a similarity with the celebrated Sinkhorn scaling algorithm, with some key differences that are discussed.

Next, we derive in Section 3.2 an efficient algorithm to compute the regularization path in $\ell_2$-penalized UOT. To do so, we build on our proposed reformulation and more precisely on the fact that $\ell_2$-penalized UOT can be reformulated as a weighted Lasso problem. We propose a new methodology inspired by LARS (Efron et al., 2004; Hastie et al., 2004), which, to the best of our knowledge, is the first regularization path algorithm for OT problems. It brings a novel understanding of the properties of the evolution of the support of OT plans, besides the practical interest of computing the complete regularization path when hyperparameter validation is necessary.

Our new families of algorithms (MM for general UOT, LARS for $\ell_2$-penalized UOT) are showcased in the numerical experiments of Section 4. Python implementation of the algorithms, provided in supplementary, will be released with MIT license on GitHub. The connection between UOT and linear regression that we reveal in the paper opens the door to further fruitful developments and in particular to more efficient algorithms, thanks to the large literature dealing with non-negative penalized linear regression. We discuss those possible research directions in Section 5, before concluding the paper.

**Notations.** Vectors such as $\boldsymbol{m}$ are written with lower case and bold font, with coefficients $m_i$ or $[\boldsymbol{m}]_i$, according to context. The $|\mathcal{A}|$-dimensional sub-vector with indexes in set $\mathcal{A}$ is written $\boldsymbol{m}_{\mathcal{A}}$. Matrices such as $\boldsymbol{M}$ are written with upper case and bold font, with coefficients $M_{i,j}$. We introduce a vectorization operator defined by $\boldsymbol{m} = \text{vec}(\boldsymbol{M}) = [M_{1,1}, M_{1,2}, \ldots, M_{n,m-1}, M_{n,m}]^\top$, i.e., the

concatenation of the *rows* of the matrix, following the Numpy/C memory convention. $\mathbb{1}_n$ is a vector of $n$ ones and $M \geq 0$ denotes entry-wise non-negativity. Finally, $D_\varphi$ is the Bregman divergence generated by the strictly convex and differentiable function $\varphi$, i.e., $D_\varphi(\boldsymbol{u}, \boldsymbol{v}) = \sum_i d_\varphi(u_i, v_i) = \sum_i [\varphi(u_i) - \varphi(v_i) - \varphi'(v_i)(u_i - v_i)]$.

## 2 Reformulation of UOT as non-negative penalized linear regression

### 2.1 Background on Optimal Transport

Let us consider two clouds of points $\boldsymbol{X} = \{\boldsymbol{x}_i\}_{i=1}^n$ and $\boldsymbol{Y} = \{\boldsymbol{y}_j\}_{j=1}^m$. Let $\boldsymbol{a} \in \mathbb{R}_n^+$ and $\boldsymbol{b} \in \mathbb{R}_m^+$ be two discrete distributions of mass on $\boldsymbol{X}$ and $\boldsymbol{Y}$, such that $a_i$ (resp. $b_j$) is the mass at $\boldsymbol{x}_i$ (resp. $\boldsymbol{y}_j$). The *balanced* OT problem, as defined by Kantorovich (1942), is a linear problem that computes the minimum cost of moving $\boldsymbol{a}$ to $\boldsymbol{b}$:

$$\mathrm{OT}(\boldsymbol{a}, \boldsymbol{b}) = \min_{\boldsymbol{T} \geq 0} \langle \boldsymbol{C}, \boldsymbol{T} \rangle \quad \text{such that (s.t.)} \quad \boldsymbol{T}\mathbb{1}_m = \boldsymbol{a}, \boldsymbol{T}^\top \mathbb{1}_n = \boldsymbol{b} \tag{1}$$

where $\langle \cdot, \cdot \rangle$ is the Frobenius inner product, $\boldsymbol{T} \in \mathbb{R}_{n \times m}^+$ is the *transport plan* and $\boldsymbol{C} \in \mathbb{R}_{n \times m}^+$ is the *cost matrix*. The entry $C_{i,j}$ of $\boldsymbol{C}$ represents the cost of moving point $\boldsymbol{x}_i$ to $\boldsymbol{y}_j$. The Wasserstein 1-distance (also known as the earth mover's distance) is obtained for $C_{i,j} = \|\boldsymbol{x}_i - \boldsymbol{y}_j\|$. The constraints on the transport plan $\boldsymbol{T}$ require that $\|\boldsymbol{a}\|_1 = \|\boldsymbol{b}\|_1$ and that *all* the mass from $\boldsymbol{a}$ is transported to $\boldsymbol{b}$. These constraints can be alleviated through relaxation, leading to UOT (Benamou, 2003):

$$\mathrm{UOT}^{\boldsymbol{\lambda}}(\boldsymbol{a}, \boldsymbol{b}) = \min_{\boldsymbol{T} \geq 0} \quad \langle \boldsymbol{C}, \boldsymbol{T} \rangle + \lambda_1 D_\varphi(\boldsymbol{T}\mathbb{1}_m, \boldsymbol{a}) + \lambda_2 D_\varphi(\boldsymbol{T}^\top \mathbb{1}_n, \boldsymbol{b}). \tag{2}$$

The deviations from the true marginals are penalized by means of a given Bregman divergence $D_\varphi$, as introduced in Chizat et al. (2018), where $\lambda_1$ and $\lambda_2$ are hyperparameters that represent the strengths of penalization. Note that, in the case of $\|\boldsymbol{a}\|_1 = \|\boldsymbol{b}\|_1$, balanced OT (Eq. 1) is recovered when $\lambda_1 = \lambda_2 \to \infty$. Furthermore, when $\lambda_1$ or $\lambda_2 \to \infty$, we recover semi-relaxed OT (Rabin et al., 2014). In practice, authors often set $\lambda_1 = \lambda_2 = \lambda$ for UOT in order to reduce the necessity of hyperparameter tuning. Various divergences have been considered in the literature. The $\ell_1$ norm gives rise to so-called *partial* optimal transport (Caffarelli and McCann, 2010). The squared $\ell_2$ norm provides a sparse and smooth transport plan (Blondel et al., 2018) when introducing a strongly convex term in Eq. (2). Chizat et al. (2018) derive efficient algorithms to solve Eq. (2) for several divergences by adding an additional regularization term $\lambda_{\mathrm{reg}} D_\varphi(\boldsymbol{T}, \boldsymbol{a}\boldsymbol{b}^\top)$. In particular, entropic regularization is obtained when the KL divergence is used, promoting a dense transport plan unlike exact UOT.

### 2.2 Reformulation of UOT

**UOT cast as regression.** Let $\boldsymbol{t} = \mathrm{vec}(\boldsymbol{T})$, $\boldsymbol{c} = \mathrm{vec}(\boldsymbol{C})$ and $\boldsymbol{y}^\top = [\boldsymbol{a}^\top, \boldsymbol{b}^\top]$. Problem (2) can be re-written as

$$\min_{\boldsymbol{t} \geq 0} \quad F_\lambda(\boldsymbol{t}) \stackrel{\text{def}}{=} \frac{1}{\lambda} \boldsymbol{c}^\top \boldsymbol{t} + D_\varphi(\boldsymbol{H}\boldsymbol{t}, \boldsymbol{y}) \tag{3}$$

and as such be expressed as a non-negative penalized linear regression problem, where the *design matrix* $\boldsymbol{H} = [\boldsymbol{H}_r^\top, \boldsymbol{H}_c^\top]^\top$ is the concatenation of the matrices $\boldsymbol{H}_r$ and $\boldsymbol{H}_c$ that compute sums of the rows and columns of $\boldsymbol{T}$, respectively (see expressions in Section **??** of the supplementary material). Note that, for the sake of simplicity, we consider here $\lambda_1 = \lambda_2 = \lambda$ but this hypothesis could be easily alleviated for a given family of divergences (see Sec. 5 for a discussion). Important features of Eq. (3) should be discussed. First, $F_\lambda(\boldsymbol{t})$ is convex thanks to the convexity of Bregman divergences w.r.t. their first argument. Second, $\boldsymbol{H}$ is very structured and sparse (with a ratio of only $\frac{1}{m+n}$ non-zero coefficients) which will allow for more efficient computations and updates than with a dense $\boldsymbol{H}$. Finally, since $\boldsymbol{t} \geq 0$ and $\boldsymbol{c} \geq 0$, the linear term can be expressed as $\frac{1}{\lambda} \boldsymbol{c}^\top \boldsymbol{t} = \frac{1}{\lambda} \sum_i c_i t_i = \frac{1}{\lambda} \sum_i c_i |t_i|$. This corresponds to a weighted $\ell_1$ regularization, promoting sparsity in $\boldsymbol{t}$ and hence in the transport plans. Note that the "sparse" regularization is here controlled by $\frac{1}{\lambda}$ (instead of $\lambda$ in classical penalized linear regression), meaning that the sparsity promoting term will be more aggressive for small $\lambda$.

**Solving problem** (3). Problems of the form of Eq. (3) are well-known in inverse problems and NMF. In inverse problems, $\boldsymbol{t}$ typically acts as a clean image degraded by operator $\boldsymbol{H}$ (e.g., a convolution) and noise. The data fitting term $D_\varphi(\boldsymbol{H}\boldsymbol{t}, \boldsymbol{y})$ captures assumptions about the noise corrupting the observed

image $\boldsymbol{y}$. Sparsity is a common regularizer of $\boldsymbol{t}$. In NMF, given a set of nonnegative samples $\{\boldsymbol{y}_l\}$ one wants to learn a non-negative dictionary $\boldsymbol{H}$ and non-negative lower-dimensional embeddings $\{\boldsymbol{t}_l\}$ such that $\boldsymbol{y}_l \approx \boldsymbol{H}\boldsymbol{t}_l$ (Lee and Seung, 1999). Updating the latter involves optimization problems of form (3) (with or without sparse regularization). In contrast to problem (3), the data fitting term is more commonly $D_\varphi(\boldsymbol{y}, \boldsymbol{H}\boldsymbol{t})$ instead of $D_\varphi(\boldsymbol{H}\boldsymbol{t}, \boldsymbol{y})$ in inverse problems and NMF. This is because the former is a log-likelihood in disguise for the mean-parametrized exponential family, and takes important noise models as special cases, such as Poisson, additive Gaussian or multiplicative Gamma noise (Févotte and Idier, 2011). Using such penalizations with reversed arguments would be possible in our case as well but we stick to the now standard formulation of (Liero et al., 2018; Chizat et al., 2018) for simplicity.

In the next section, we will first leverage a classical family of algorithms in inverse problems and NMF, namely MM, to obtain new algorithms for KL and $\ell_2$-penalized UOT (possibly with entropic regularization in the first case). Second, we will leverage results about non-negative Lasso to design an efficient algorithm to compute the regularization path of $\ell_2$-penalized UOT.

## 3 Novel numerical solvers for UOT

### 3.1 Majorization-Minimization (MM) for UOT

**General MM framework.** MM algorithms have been around a long time in inverse problems and NMF to solve problems of form (3). Classical algorithms for NMF such as (Lee and Seung, 2001) have built on seminal MM algorithms for inverse problems such as (Richardson, 1972; De Pierro, 1993). Subsequent works in NMF such as (Dhillon and Sra, 2005; Févotte and Idier, 2011; Yang and Oja, 2011) have further contributed novel MM algorithms for larger classes of problems, including larger families of divergences. In a nutshell, MM consists in iteratively building and minimizing an upper bound of the objective function which is tight at the current parameter estimate (and referred to as *auxiliary function*), see Hunter and Lange (2004); Sun et al. (2017) for tutorials. In NMF, a common approach consists of alternating the updates of the dictionary $\boldsymbol{H}$ and of the embeddings. In our case, $\boldsymbol{H}$ is fixed and we may use the results of (Dhillon and Sra, 2005) to build an auxiliary function for term $D_\varphi(\boldsymbol{H}\boldsymbol{t}, \boldsymbol{y})$, to which we may simply add the linear term $\boldsymbol{c}^\top \boldsymbol{t}/\lambda$ to obtain a valid auxiliary function for $F_\lambda(\boldsymbol{t})$. Let $\tilde{\boldsymbol{t}}$ denote the current estimate of $\boldsymbol{t}$, $\tilde{Z}_{i,j} = \frac{H_{i,j}\tilde{t}_j}{\sum_l H_{i,l}\tilde{t}_l}$ and

$$G_\lambda(\boldsymbol{t}, \tilde{\boldsymbol{t}}) = \sum_{i,j} \tilde{Z}_{i,j}\varphi\left(\frac{H_{i,j}t_j}{\tilde{Z}_{i,j}}\right) + \sum_j \left[\frac{c_j}{\lambda} - \sum_i H_{i,j}\varphi'(y_i)\right] t_j + cst, \qquad (4)$$

where $cst = \sum_i[\varphi'(y_i)y_i - \varphi(y_i)]$. Then, $G_\lambda(\boldsymbol{t}, \tilde{\boldsymbol{t}})$ is an auxiliary function for $F_\lambda(\boldsymbol{t})$, i.e., $\forall \boldsymbol{t}$, $G_\lambda(\boldsymbol{t}, \tilde{\boldsymbol{t}}) \geq F_\lambda(\boldsymbol{t})$ and $G_\lambda(\tilde{\boldsymbol{t}}, \tilde{\boldsymbol{t}}) = F_\lambda(\tilde{\boldsymbol{t}})$. Let $\boldsymbol{t}^{(k+1)} = \operatorname{argmin}_{\boldsymbol{t} \geq 0} G_\lambda(\boldsymbol{t}, \boldsymbol{t}^{(k)})$, then $F_\lambda(\boldsymbol{t}^{(k)}) = G_\lambda(\boldsymbol{t}^{(k)}, \boldsymbol{t}^{(k)}) \geq G_\lambda(\boldsymbol{t}^{(k+1)}, \boldsymbol{t}^{(k)}) \geq F_\lambda(\boldsymbol{t}^{(k+1)})$, producing a descent algorithm over $F$. The trick to obtain $G$ is to apply Jensen inequality to $\varphi(\sum_j H_{i,j}t_j) = \varphi(\sum_j \tilde{Z}_{i,j}\frac{H_{i,j}}{\tilde{Z}_{i,j}}t_j) \leq \sum_j \tilde{Z}_{i,j}\varphi(\frac{H_{i,j}}{\tilde{Z}_{i,j}}t_j)$, thanks to the convexity of $\varphi$, see details in (Dhillon and Sra, 2005). We provide below the resulting algorithms for the KL and $\ell_2$ penalizations, with detailed computations available in Section **??** of the supplementary.

**MM for KL-penalized UOT.** The KL divergence is obtained with $\varphi(y) = y \log y - y$. Minimizing $G_\lambda(\boldsymbol{t}, \boldsymbol{t}^{(k)})$ in that case leads to following multiplicative update:

$$t_j^{(k+1)} = t_j^{(k)} \exp\left(\frac{[\boldsymbol{H}^\top \log(\boldsymbol{y}) - \boldsymbol{H}^\top \log(\boldsymbol{H}\boldsymbol{t}^{(k)})]_j - \frac{1}{\lambda}c_j}{[\boldsymbol{H}^\top \mathbb{1}]_j}\right). \qquad (5)$$

Owing to the structure of $\boldsymbol{t}$ and $\boldsymbol{H}$, the update can be re-written in the following matrix form:

$$\boldsymbol{T}^{(k+1)} = \operatorname{diag}\left(\frac{\boldsymbol{a}}{\boldsymbol{T}^{(k)}\mathbb{1}_m}\right)^{\frac{1}{2}} \left(\boldsymbol{T}^{(k)} \odot \exp\left(-\frac{\boldsymbol{C}}{2\lambda}\right)\right) \operatorname{diag}\left(\frac{\boldsymbol{b}}{\boldsymbol{T}^{(k)\top}\mathbb{1}_n}\right)^{\frac{1}{2}}, \qquad (6)$$

where $\odot$ is entrywise multiplication and divisions are taken entrywise as well. The multiplicative update (6) is remarkably similar to the well-known Sinkhorn-Knopp algorithm that has been used in

numerous OT problems involving KL regularization. But instead of two separate steps for the left and right scaling, Eq. (6) applies these scalings simultaneously in a unique update using the diagonal matrices (and a form of geometrical average). Also note how the scaling factor $\exp\left(-\frac{C}{2\lambda}\right)$ penalizes along iterations the coefficients of the transport plan with large costs.

**MM for $\ell_2$-penalized UOT.** The quadratic loss is obtained with $\varphi(y) = \frac{y^2}{2}$. In that case, minimizing $G_\lambda(t, t^{(k)})$ s.t. non-negativity leads to following multiplicative update:

$$T^{(k+1)} = T^{(k)} \odot \frac{\max\left(0, a\mathbb{1}_m^\top + \mathbb{1}_n b^\top - \frac{1}{\lambda}C\right)}{T^{(k)}O_m + O_n T^{(k)}} \quad \text{with} \quad O_\ell = \mathbb{1}_\ell \mathbb{1}_\ell^\top. \tag{7}$$

Interestingly enough, update (7) prunes any coefficient $T_{i,j}$ in $T$ such that $a_i + b_j - \frac{1}{\lambda}C_{i,j} < 0$ from the very first iteration, providing a useful certificate on the support of the solution.

## 3.2 Regularization path for $\ell_2$-penalized UOT

Let us focus on the case where $D_\varphi$ is a quadratic divergence. As mentioned in Section 2.2, Eq. (3) is then a positive weighted Lasso problem, allowing us to derive the first regularization path algorithm for computing the whole set of solutions for a varying $\lambda$ from 0 to $+\infty$. Note that the path's extreme point recovers the balanced OT solution. We show that the path is piecewise linear in $1/\lambda$ between changes in the active set $\mathcal{A} = \text{supp}(t^\lambda)$, where $t^\lambda = \text{vec}(T^\lambda)$ and $T^\lambda$ is the OT plan for given hyperparameter $\lambda$. The main steps of the algorithm are roughly as follows: given a current solution $(\lambda_k, T^{\lambda_k})$ and a current active set $\mathcal{A}_k$, we look for the next value $\lambda_{k+1} > \lambda_k$ such that the active set changes (i.e., $\mathcal{A}_{k+1} \neq \mathcal{A}_k$), either because one component enters or leaves the active set. We describe our algorithm below.

**KKT conditions of the $\ell_2$-penalized UOT problem.** The Lagrangian for problem (3) writes:

$$L_\lambda(t, \gamma) = \frac{1}{\lambda}c^\top t + \frac{1}{2}(Ht - y)^\top(Ht - y) - \gamma^\top t \tag{8}$$

where $\gamma$ represents the Lagrange parameters. We denote $m = H^\top y = \text{vec}(a\mathbb{1}_m^\top + \mathbb{1}_n b^\top)$. KKT optimality conditions state that i) $\nabla_t L_\lambda(t, \lambda) = \frac{1}{\lambda}c + H^\top H t - m - \gamma = 0$ (stationarity condition), ii) $\gamma \odot t = 0$ (complementary condition) and iii) $\gamma \geq 0$ (feasibility condition).

**Piecewise linearity of the path.** Assume that, at iteration $k$, we know the current active set $\mathcal{A} = \mathcal{A}_k$ and we look for $t_\mathcal{A}^\lambda$ (the other values of $t_\mathcal{A}$ being 0). Let $H_\mathcal{A}$, $m_A$ and $c_\mathcal{A}$ denote the corresponding sub-matrix and vectors (see Appendix **??** for rigorous definitions). Because of the complementary condition, we have $\gamma_\mathcal{A} = 0$. Using $\lambda = \lambda_k + \epsilon$, with $\epsilon > 0$ small enough to ensure that the active set remains the same, the stationarity condition writes:

$$H_\mathcal{A}^\top H_\mathcal{A} t_\mathcal{A}^\lambda = m_\mathcal{A} - \frac{1}{\lambda}c_\mathcal{A} \quad \Rightarrow \quad t_\mathcal{A}^\lambda = \tilde{m}_\mathcal{A} - \frac{1}{\lambda}\tilde{c}_\mathcal{A} \tag{9}$$

with $\tilde{m}_\mathcal{A} = (H_\mathcal{A}^\top H_\mathcal{A})^{-1} m_\mathcal{A}$ and $\tilde{c}_\mathcal{A} = (H_\mathcal{A}^\top H_\mathcal{A})^{-1} c_\mathcal{A}$. Eq. (9) shows that the optimal $t_\mathcal{A}^\lambda$ (and hence $t^\lambda$) can be solved for any $\lambda \in [\lambda_k, \lambda_{k+1}]$, i.e., when the active set $\mathcal{A}$ remains the same, by solving a linear problem. It also reveals the piecewise linearity in $\lambda^{-1}$ of the path when $\mathcal{A}$ is fixed. As expected, balanced OT is recovered when $\lambda \to \infty$.

**Finding $(\lambda_{k+1}, \mathcal{A}_{k+1})$ given $(\lambda_k, \mathcal{A}_k)$.** Given a current solution $(\lambda_k, t^{\lambda_k})$ and $\lambda = \lambda_k + \epsilon$, we increase the $\epsilon$ until we reach a change in the set of active components. This happens whenever the first of the following two situations occurs.

● **One component in $\mathcal{A}$ becomes inactive.** In that case, we remove the index $i \in \mathcal{A}$ with the smallest $\lambda_r > \lambda_k$ that violates the constraint. In such case, $[\tilde{m}_\mathcal{A}]_i = [\tilde{c}_\mathcal{A}]_i/\lambda$ and we may write

$$\lambda_r = \min_{>\lambda_k}\left(\frac{\tilde{c}_\mathcal{A}}{\tilde{m}_\mathcal{A}}\right) \tag{10}$$

where $\min_{>\lambda_k}$ indicates the minimum value in the vector greater than $\lambda_k$ and the division is entrywise.

**Algorithm 1** Regularization path of $\ell_2$-penalized UOT

---

**Require:** $\boldsymbol{a}, \boldsymbol{b}, \boldsymbol{C}, \lambda_0 = 0, \boldsymbol{t}_0 = \boldsymbol{0}, \mathcal{A} = \mathcal{A}_0 = \emptyset, k = 1$

$\lambda_1 = \min \dfrac{\boldsymbol{c}_{\bar{\mathcal{A}}}}{\boldsymbol{m}_{\bar{\mathcal{A}}}}, \mathcal{A} = \mathcal{A}_1 = \arg\min \dfrac{\boldsymbol{c}_{\bar{\mathcal{A}}}}{\boldsymbol{m}_{\bar{\mathcal{A}}}}, \boldsymbol{H}_{\mathcal{A}}^{\top}\boldsymbol{H}_{\mathcal{A}} = 2$

$\boldsymbol{t}_{\mathcal{A}_1}^{\lambda_1} = \dfrac{\boldsymbol{m}_{\mathcal{A}}}{2} - \dfrac{1}{\lambda_1}\dfrac{\boldsymbol{c}_{\mathcal{A}}}{2}$

**while** $(\boldsymbol{H}\boldsymbol{t}^{\lambda_k} - \boldsymbol{y})^{\top}(\boldsymbol{H}\boldsymbol{t}^{\lambda_k} - \boldsymbol{y}) \neq 0$ **do**

  $\lambda_r, \lambda_a \leftarrow$ Compute as in Eq. (10) and Eq. (11)

  $\lambda_{k+1} \leftarrow \min(\lambda_r, \lambda_a)$

  $\boldsymbol{t}_{\mathcal{A}}^{\lambda_{k+1}} \leftarrow (\boldsymbol{H}_{\mathcal{A}}^{\top}\boldsymbol{H}_{\mathcal{A}})^{-1}\boldsymbol{m}_{\mathcal{A}} - \dfrac{1}{\lambda_{k+1}}(\boldsymbol{H}_{\mathcal{A}}^{\top}\boldsymbol{H}_{\mathcal{A}})^{-1}\boldsymbol{c}_{\mathcal{A}}$

  $\mathcal{A} = \mathcal{A}_{k+1} \leftarrow$ Update active set for next iteration.

  $(\boldsymbol{H}_{\mathcal{A}}^{\top}\boldsymbol{H}_{\mathcal{A}})^{-1} \leftarrow$ Update from $(\boldsymbol{H}_{\mathcal{A}_k}^{\top}\boldsymbol{H}_{\mathcal{A}_k})^{-1}$ with Schur complement (see supplementary **??**)

  $k \leftarrow k + 1$

**end while**

**return** $(\lambda_k, \boldsymbol{t}^{\lambda_k})_k$

---

• **One component in $\bar{\mathcal{A}}$ becomes active.** This occurs when the KKT positivity constraint $\boldsymbol{\gamma}_{\bar{\mathcal{A}}} \geq \boldsymbol{0}$ becomes violated. Assume this happens at index $i \in \bar{\mathcal{A}}$ for the smallest value $\lambda_a > \lambda_k$ of $\lambda$. In such case, the stationarity condition outside the active set can be rewritten:

$$\left[\frac{1}{\lambda}\boldsymbol{c}_{\bar{\mathcal{A}}} + \left[\boldsymbol{H}^{\top}\boldsymbol{H}\left(\tilde{\boldsymbol{m}} + \frac{1}{\lambda}\tilde{\boldsymbol{c}}\right)\right]_{\bar{\mathcal{A}}} - \boldsymbol{m}_{\bar{\mathcal{A}}}\right]_i = [\boldsymbol{\gamma}_{\bar{\mathcal{A}}}]_i \Rightarrow \lambda_a = \min_{>\lambda_k}\left(\frac{\boldsymbol{c}_{\bar{\mathcal{A}}} - \left[\boldsymbol{H}^{\top}\boldsymbol{H}\tilde{\boldsymbol{c}}\right]_{\bar{\mathcal{A}}}}{\boldsymbol{m}_{\bar{\mathcal{A}}} - \left[\boldsymbol{H}^{\top}\boldsymbol{H}\tilde{\boldsymbol{m}}\right]_{\bar{\mathcal{A}}}}\right), \quad (11)$$

where $\tilde{\boldsymbol{m}}$ (resp. $\tilde{\boldsymbol{c}}$) equals $\tilde{\boldsymbol{m}}_{\mathcal{A}}$ (resp. $\tilde{\boldsymbol{c}}_{\mathcal{A}}$) on $\mathcal{A}$ and zero on $\bar{\mathcal{A}}$.

In practice, at each step of the path, we compute both $\lambda_r$ and $\lambda_a$ and set $\lambda_{k+1} = \min\{\lambda_r, \lambda_a\}$. The active set $\mathcal{A}_{k+1}$ is obtained by either removing the index $i \in \mathcal{A}$ correponding of the $\arg\min$ of eq. (10) (case $\lambda_{k+1} = \lambda_r$) or by adding the index $i \in \bar{\mathcal{A}}$ corresponding to the $\arg\min$ of eq. (11) (case $\lambda_{k+1} = \lambda_a$).

**Numerical computation of the entire path.** Eq. (9) involves the computation of the matrix $(\boldsymbol{H}_{\mathcal{A}}^{\top}\boldsymbol{H}_{\mathcal{A}})^{-1}$, which is of size $|\mathcal{A}| \times |\mathcal{A}|$. As only one index leaves or enters the active set at each iteration, we can use the Schur complement of the matrix to compute its value from $(\boldsymbol{H}_{\mathcal{A}_k}^{\top}\boldsymbol{H}_{\mathcal{A}_k})^{-1}$, alleviating the computational burden of the algorithm as it only involves matrix-vector computations (see Section **??** of supplementary). Algorithm 1 sums up the different steps of the full path computation. At each iteration, we compute $\lambda_a, \lambda_r$, update the inverse matrix $(\boldsymbol{H}_{\mathcal{A}_k}^{\top}\boldsymbol{H}_{\mathcal{A}_k})^{-1}$ and estimate the solution $\boldsymbol{t}^{\lambda_{k+1}}$ with a complexity of $O(nm)$.

**Regularization path of the semi-relaxed $\ell_2$-penalized UOT.** As a side result, let us consider the semi-relaxed OT problem $\text{SROT}^{\lambda}(\boldsymbol{a}, \boldsymbol{b}) = \min_{\boldsymbol{T} \geq 0, \boldsymbol{T}^{\top}\mathbb{1}_n = \boldsymbol{b}}\langle \boldsymbol{C}, \boldsymbol{T}\rangle + \lambda\|\boldsymbol{T}\mathbb{1}_m - \boldsymbol{a}\|^2$. The main difference with UOT is that the equality constraint $\boldsymbol{T}^{\top}\mathbb{1}_n = \boldsymbol{b}$ (equivalent to $\boldsymbol{H}_c\boldsymbol{t} = \boldsymbol{b}$) must always be met. This leads to the following Lagrangian:

$$L_{\lambda}(\boldsymbol{t}, \boldsymbol{\gamma}, \boldsymbol{u}) = \frac{1}{\lambda}\boldsymbol{c}^{\top}\boldsymbol{t} + \frac{1}{2}(\boldsymbol{H}_r\boldsymbol{t} - \boldsymbol{a})^{\top}(\boldsymbol{H}_r\boldsymbol{t} - \boldsymbol{a}) + (\boldsymbol{H}_c\boldsymbol{t} - \boldsymbol{b})^{\top}\boldsymbol{u} - \boldsymbol{\gamma}^{\top}\boldsymbol{t}, \quad (12)$$

where $\boldsymbol{u} \in \mathbb{R}^m$ contains the Lagrange parameters associated to the $m$ equality constraints. The KKT optimality conditions now dictate that i) $\nabla_{\boldsymbol{t}}L_{\lambda}(\boldsymbol{t}, \boldsymbol{\gamma}, \boldsymbol{u}) = \frac{1}{\lambda}\boldsymbol{c} + \boldsymbol{H}_r^{\top}\boldsymbol{H}_r\boldsymbol{t} - \boldsymbol{H}_r^{\top}\boldsymbol{a} + \boldsymbol{H}_c^{\top}\boldsymbol{u} - \boldsymbol{\gamma} = 0$, ii) $\boldsymbol{\gamma} \odot \boldsymbol{t} = 0$, iii) $\boldsymbol{\gamma} \geq 0$ and $\boldsymbol{H}_c\boldsymbol{t} - \boldsymbol{b} = \boldsymbol{0}$. We can use the same reasoning than previously to compute the entire path. Details are provided in Section **??** of the supplementary. The main difference lies in solving, at each iteration, a linear system of size $(m + |\mathcal{A}|)$ to comply with the marginal equality constraint. The path is initialized as follows: the $j^{th}$ column of $\boldsymbol{T}^0$ for $\lambda_0 = 0$ is set to the weighted canonical vector $b_{i^{\star}}\mathbf{e}_{i^{\star}}$, where $i^{\star} = \text{argmin}\{C_{i,j}\}_i$.

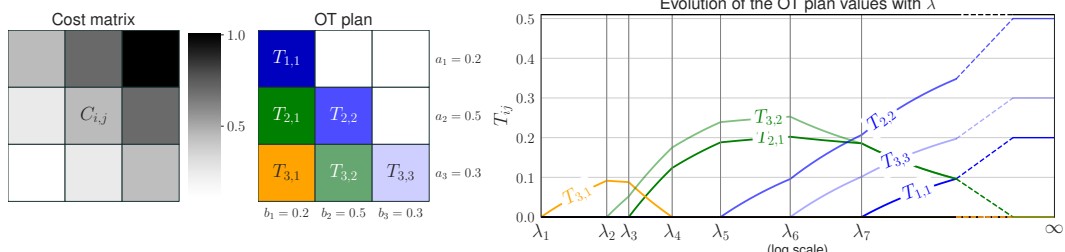

Figure 1: (Left) cost matrix $C$ (the higher the cost, the darker the color); (middle) OT plan whose cells are color-coded with respect to the $\lambda$ values at which they are activated. The blank cells never enter the active set as the corresponding cost it too high; (right) evolution of $T_{i,j}$ when $\lambda$ increases. Note that the $x$-axis is in log scale and is discontinued (but still monotonic) between $\lambda_7$ and $\infty$.

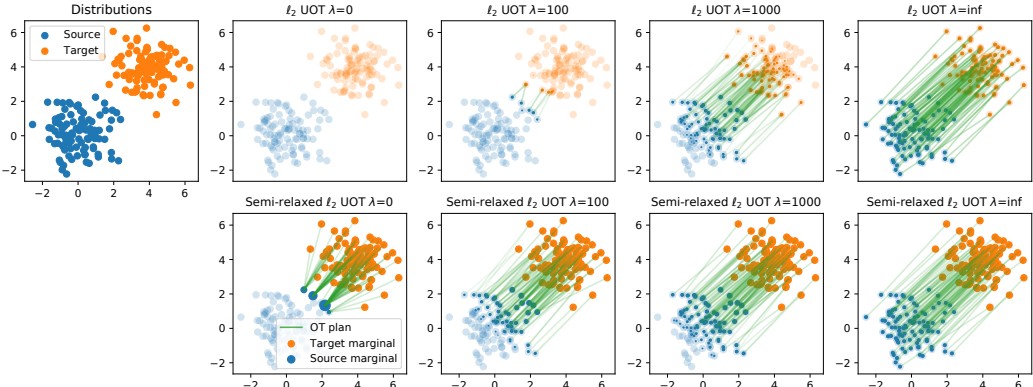

Figure 2: Regularization paths for 2D empirical distributions for $\ell_2$-penalized UOT (top) and semi-relaxed UOT (bottom). The OT plan is shown as green lines between the source and target samples when $T_{i,j} > 0$ and the resulting marginals are shown as filled circles.

## 4   Numerical experiments

In this section, we first show the solutions obtained with our solvers[2] on simple and interpretable examples. We then evaluate the computational complexity of the different algorithms and finally we show an application where the regularization path can be used on a domain adaptation problem.

**Illustration of the algorithms.**   We first illustrate the regularization path for $\ell_2$-penalized UOT on a simple example between two distributions containing 3 points each, with different masses and a cost matrix $C$ given in Fig. 1 (left). We can see on Fig. 1 (right) that, starting from $\lambda_0 = 0$ and $T = 0$, we successively add or remove components in the active set $\mathcal{A}$ when increasing the $\lambda$ values. When $\lambda = \infty$, we recover the balanced OT solution. Recall that the path is linear in $1/\lambda$ (and not $\lambda$). We then illustrate the path for both $\ell_2$-penalized UOT and semi-relaxed UOT on two 2D distributions with $n = m = 100$ samples. We can see in Fig. 2 the difference between the two regularization paths for specific values of $\lambda$. UOT starts with an empty plan for $\lambda = 0$ and then activates samples from both source and target from the closest to the farthest ones until convergence to the balanced OT plan. Semi-relaxed UOT starts with all target samples active due to marginal constraints and progressively activates the source samples.

**Comparison of the performances of the algorithms.**   We now provide an empirical evaluation of the running times of the proposed algorithms, using 2 sets of 10-dimensional points with $n = m$ and drawn according to IID Gaussian distributions. The cost matrix $C$ is computed using a squared $\ell_2$

---

[2]Our implementation of the regularization path has been contributed to POT Flamary et al. (2021) and the MM algorithms are provided in the repository `https://github.com/lchapel/UOT-though-penalized-linear-regression`.

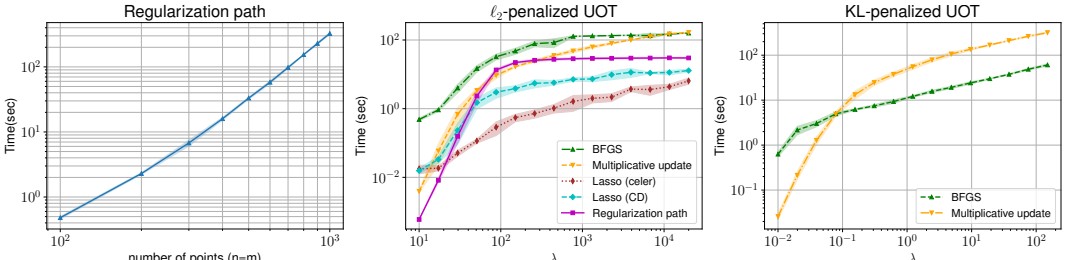

Figure 3: (Left) Running times of Alg. 1 w.r.t. the number of points; (middle) comparison of $\ell_2$-penalized UOT with $m = n = 500$ (right) likewise for KL-penalized UOT. Dark curves (resp. shaded regions) represent average (resp. variance) values over 5 runs.

norm. We first study the running times of the regularization path algorithm, for $n = m$ ranging from 100 to 1000, averaging the results over 5 runs, see Fig. 3 (left). We empirically observe that log-log plot is near-linear, with an empirical complexity $O(n^{3.27})$ in this example.

Using $n = m = 500$, we compare the running times of the current state-of-the-art BFGS algorithm (Blondel et al., 2018)[3] using SciPy (Virtanen et al., 2020) and those of our algorithms: the $\ell_2$-penalized UOT formulated as a Lasso problem (with both the Celer algorithm (Massias et al., 2018) and the coordinate descent solvers from Scikit-learn (Pedregosa et al., 2011)), the multiplicative algorithm for both the $\ell_2$ and the KL penalties and the regularization path algorithm (see Section **??** of the supplemental material for more details about the solvers and their parameters, together with a comparison of the results of the MM algorithms computed on both CPU and GPU). Figure 3 (middle and right) shows the average running time for all algorithms. For $\ell_2$-penalized UOT, we observe that, for large $\lambda$ values, the Lasso solvers are the fastest and that, whatever the value $\lambda$, BFGS is the slowest. We also notice that, for large $\lambda$, the running times for computing the path remain constant: when the last active set is found, computing the OT plan only involves a weighted sum. As for KL-penalized UOT, the BFGS algorithm is more efficient when large values of $\lambda$ are considered. One can also notice that, similarly to Sinkhorn which is fast for large regularization values, the multiplicative algorithms for both penalties are also fast for high $1/\lambda$ values.

**Regularization path for unbalanced domain adaptation.** We demonstrate the interest of having the entire regularization path in a classification context where some of the data collection may be polluted by outliers. We consider a setup similar to Mukherjee et al. (2020). Let the source $X$ be a set of 400 MNIST digits sampled from the digits $0, 1, 2, 3$ (100 points per class) and let the target $Y$ be a set of digits $0, 1$ of MNIST (LeCun et al., 2010) and of digits $8, 9$ from Fashion MNIST (Xiao et al., 2017). Our setting is simple classification: we classify a sample of the target dataset by propagating the label of the source sample it is the most transported to, provided that the transported mass of the target point is greater than $0.25b_j$. Note that similarly to Mukherjee et al. (2020) a validation set can be used here to select the best $\lambda$. Figure 4 shows the overall accuracy, defined as the number of samples that are correctly classified divided by the total number of points, and the current accuracy, which is the proportion of well-classified points among the points that are classified, i.e., that are receiving mass. One can notice that, as the number of classified points increases (with $\lambda$), the overall accuracy increases as more and more points are well classified while the current accuracy remains stable until outliers are included in the labeled set. This suggests that UOT can be used not only for classification but also as an automated outlier detection method.

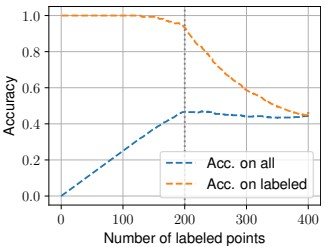

Figure 4: Evolution of the classification accuracy for the domain adaptation problem w.r.t. the number of classified points.

---

[3]Note that we cannot compare our result with solvers for entropic formulation of UOT with $\lambda_{reg} \to 0$ (that should provide a sparse transport plan) as in that case, their algorithm becomes a fixed point method that cannot converge to a solution of the problem.

# 5 Discussion and perspectives

We showed that UOT can be recast as a non-negative penalized linear regression problem, encouraging us to dig into this well-established field of research in order to adapt existing algorithmic solutions to the structure of the UOT problem. In this section, we discuss the relation between the proposed algorithms and classical solvers used in OT, and also investigate some research directions that can widen the scope of proposed methods.

**Multiplicative algorithms for UOT.** As discussed in Section 3.1, the multiplicative updates for the KL divergence obtained from MM resemble the Sinkhorn algorithm from Chizat et al. (2018), except for the joint scaling and the weighting matrix $\exp(-\boldsymbol{C}/2)$. Interestingly, this scaling matrix also appears in the Inexact Proximal Point OT (IPOT) algorithm of Xie et al. (2020) to solve balanced OT. As a matter of fact, we show in Section **??** of the supplementary that IPOT is a MM algorithm. The idea is to re-write the OT objective as $[\langle \boldsymbol{C}, \boldsymbol{T} \rangle + \lambda D_\varphi(\boldsymbol{T}, \boldsymbol{ab}^\top)] - \lambda D_\varphi(\boldsymbol{T}, \boldsymbol{ab}^\top)$ and upper bound the concave term by its tangent. This further supports the interest of MM for OT and UOT, and highlight an important feature of one of our contributions: designing the first Sinkhorn-like multiplicative algorithm for UOT that can be applied when the OT plan is not entropy-regularized.

**More efficient solvers.** Despite the positive experimental results of Section 4, multiplicative and regularization path algorithms can be slow, especially for large values of $\lambda$. Various accelerations can be envisaged. Regarding path algorithms, the approach of Mairal and Yu (2012) can compute a regularization path with precision $\epsilon$ in $o(1/\epsilon)$ iterations. This would lead in our setting to a full complexity of $O(mn/\epsilon)$ that is even interesting to approximate balanced OT. Another way to speed up computations is to use *screening*. In sparse regression, this consists of eliminating during optimization components that will not belong to the support of the solutions thanks to safe screening tests. Methods such as (El Ghaoui et al., 2012; Wang et al., 2015; Dantas et al., 2021) can readily be adapted to our $\ell_2$ or KL-penalized UOT algorithms. Finally, an other line of improvement is to consider stochastic optimization methods such as (Defazio et al., 2014). Given the particular structure of $\boldsymbol{H}$, the complexity of stochastic updates shall be small and can lead to very efficient implementations (Nesterov, 2014).

**General case and entropy-regularized UOT.** Following (Frogner et al., 2015; Chizat et al., 2018; Séjourné et al., 2019), general regularized UOT can be expressed as:

$$\text{RUOT}^{\boldsymbol{\lambda}}(\boldsymbol{a}, \boldsymbol{b}) = \min_{\boldsymbol{T} \geq 0} \quad \langle \boldsymbol{C}, \boldsymbol{T} \rangle + \lambda_1 D_\varphi(\boldsymbol{T}\mathbb{1}_m, \boldsymbol{a}) + \lambda_2 D_\varphi(\boldsymbol{T}^\top \mathbb{1}_n, \boldsymbol{b}) + \lambda_{\text{reg}} D_\varphi(\boldsymbol{T}, \boldsymbol{ab}^\top). \quad (13)$$

As it turns out, this general problem involving different regularization weights $(\lambda_1, \lambda_2, \lambda_{\text{reg}})$ can easily be addressed in our framework as well using two simple tricks. The first one consists of absorbing the regularization weights into the divergences. Indeed, many divergences are homogeneous, i.e., satisfy a relation of the form $\lambda D_\varphi(\mathbf{x}|\mathbf{y}) = D_\varphi(\lambda^\alpha \mathbf{x}|\lambda^\alpha \mathbf{y})$ where $\alpha$ is divergence-specific. This holds in particular for the KL divergence ($\alpha = 1$) and the squared $\ell_2$ norm ($\alpha = 1/2$). The second one consists of complementing $\boldsymbol{H}$ and $\boldsymbol{y}$ with suitable terms to account for the regularization term. In the end, we may re-write Eq. (13) into Eq. (3) with $\lambda = 1$, $\boldsymbol{H} = [\lambda_1^\alpha \boldsymbol{H}_r^\top, \lambda_2^\alpha \boldsymbol{H}_c^\top, \lambda_{\text{reg}}^\alpha \mathbf{I}]^\top$ and $\boldsymbol{y}^\top = [\lambda_1^\alpha \boldsymbol{a}^\top, \lambda_2^\alpha \boldsymbol{b}^\top, \lambda_{\text{reg}}^\alpha \text{vec}(\boldsymbol{ab}^\top)^\top]$. In particular, we obtain the following multiplicative update in the case of entropy-regularized KL-penalized UOT:

$$\boldsymbol{T}^{(k+1)} = \text{diag}\left(\frac{\boldsymbol{a}}{\boldsymbol{T}^{(k)}\mathbb{1}_m}\right)^{\frac{\lambda_1}{\lambda_{\text{all}}}} \left(\left(\boldsymbol{T}^{(k)}\right)^{\frac{\lambda_1 + \lambda_2}{\lambda_{\text{all}}}} \odot \boldsymbol{K}\right) \text{diag}\left(\frac{\boldsymbol{b}}{\boldsymbol{T}^{(k)\top}\mathbb{1}_n}\right)^{\frac{\lambda_2}{\lambda_{\text{all}}}} \quad (14)$$

where $\boldsymbol{K} = \left(\boldsymbol{ab}^\top\right)^{\frac{\lambda_{\text{reg}}}{\lambda_{\text{all}}}} \odot \exp\left(-\frac{1}{\lambda_{\text{all}}}\boldsymbol{C}\right)$ and $\lambda_{\text{all}} = \lambda_1 + \lambda_2 + \lambda_{\text{reg}}$. This multiplicative update is slightly more complex than the Sinkhorn algorithms of Frogner et al. (2015); Chizat et al. (2018) and as such, it might have limited practical interest but is conceptually interesting and novel. Note that balanced UOT as of Eq. (2) is simply obtained with $\lambda_{\text{reg}} = 0$.

**Non-linear UOT.** Finally, we discuss how our proposed reformulation of UOT can accommodate non-linear variants in which the linear term $\langle \boldsymbol{C}, \boldsymbol{T} \rangle$ is replaced by a sparsity/robustness-promoting

term, leading to problems of the form

$$\text{NLUOT}^{\boldsymbol{\lambda}}(\boldsymbol{a}, \boldsymbol{b}) = \min_{\boldsymbol{T} \geq 0} \sum_{i,j} g(C_{i,j} T_{i,j}) + \lambda_1 D_\varphi(\boldsymbol{T} \mathbb{1}_m, \boldsymbol{a}) + \lambda_2 D_\varphi(\boldsymbol{T}^\top \mathbb{1}_n, \boldsymbol{b}) \qquad (15)$$

where $g(\cdot)$ is a usually concave function, see, e.g., (Candes et al., 2008; Gasso et al., 2009). Our MM setting can readily accommodate such a formulation by majorizing the concave terms by their tangent. The non-linearity may improve robustness w.r.t outliers and better model realistic OT problems. For instance, in real life, the costs of transporting some goods between two places can be nonlinear due to economies of scale.

**Broad and potential negative societal impact.** The contributions in this paper are methodological and focus on a reformulation of a fundamental OT problem and adapting existing algorithms to solve it. In this sense, we bring more efficient solvers that run on GPU but this computational advantage can be counterbalanced by the possibility that it brings to be applied on larger datasets. The application of OT in domain adaptation has shown that it can be used to infer labels on samples/individuals when no labels are available, suggesting a capacity for violating user privacy. A potential application of UOT is the case where two datasets of users acquired by different methods contain some shared users. UOT can be used here to find correspondences between the users in the two datasets and also identify unique users in each dataset (those that do not receive mass).

## 6 Conclusion

In this paper, we reformulate the UOT problem as a non-negative penalized linear regression, allowing us to propose two new classes of algorithms. We first derive multiplicative algorithms for both KL and $\ell_2$-penalized UOT, providing numerical solutions that are fast and easy to implement. For the specific case of $\ell_2$-penalized UOT, we provide the first regularization path algorithm that computes the whole set of solutions for *all* the regularization parameter values. We finally build on the extensive literature in inverse problem and NMF to draw some fruitful perspectives on even more efficient algorithmic solutions or the definition of new OT problems.

## Acknowledgments and Disclosure of Funding

The authors want to thank Hicham Janati for interesting discussions and providing us with the experiments of convergence for the MM algorithm in the supplemental. This work is partially funded by the French National Research Agency (ANR; grants OATMIL ANR-17-CE23-0012, RAIMO ANR-20-CHIA-0021-01, MULTISCALE ANR-18-CE23-0022-01, E4C ANR-18-EUR-0006-02, 3IA Côte d'Azur ANR-19-P3IA-0002, 3IA ANITI ANR-19-PI3A-0004) and the European Research Council (ERC; grant FACTORY-CoG-6681839). Furthermore, this research was produced within the framework of Energy4Climate Interdisciplinary Center (E4C) of IP Paris and Ecole des Ponts ParisTech. This action benefited from the support of the Chair "Challenging Technology for Responsible Energy" led by l'X - Ecole Polytechnique and the Fondation de l'Ecole Polytechnique, sponsored by TOTAL.

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
