_{\mathrm{reg}}}{\lambda_{\mathrm{all}}}} \odot \exp\left(-\frac{1}{\lambda_{\mathrm{all}}} \boldsymbol{C}\right)$ and $\lambda_{\mathrm{all}} = \lambda_1 + \lambda_2 + \lambda_{\mathrm{reg}}$. This multiplicative update is slightly more complex than the Sinkhorn algorithms of Frogner et al. (2015); Chizat et al. (2018) and as such, it might have limited practical interest but is conceptually interesting and novel. Note that balanced UOT as of Eq. (2) is simply obtained with $\lambda_{\mathrm{reg}} = 0$.

**Non-linear UOT.**   Finally, we discuss how our proposed reformulation of UOT can accommodate non-linear variants in which the linear term $\langle \boldsymbol{C}, \boldsymbol{T} \rangle$ is replaced by a sparsity/robustness-promoting

term, leading to problems of the form

$$\text{NLUOT}^{\boldsymbol{\lambda}}(\boldsymbol{a}, \boldsymbol{b}) = \min_{\boldsymbol{T} \geq 0} \quad \sum_{i,j} g(C_{i,j} T_{i,j}) + \lambda_1 D_\varphi(\boldsymbol{T} \mathbb{1}_m, \boldsymbol{a}) + \lambda_2 D_\varphi(\boldsymbol{T}^\top \mathbb{1}_n, \boldsymbol{b}) \qquad (15)$$

where $g(\cdot)$ is a usually concave function, see, e.g., (Candes et al., 2008; Gasso et al., 2009). Our MM setting can readily accommodate such a formulation by majorizing the concave terms by their tangent. The non-linearity may improve robustness w.r.t outliers and better model realistic OT problems. For instance, in real life, the costs of transporting some goods between two places can be nonlinear due to economies of scale.

**Broad and potential negative societal impact.** The contributions in this paper are methodological and focus on a reformulation of a fundamental OT problem and adapting existing algorithms to solve it. In this sense, we bring more efficient solvers that run on GPU but this computational advantage can be counterbalanced by the possibility that it brings to be applied on larger datasets. The application of OT in domain adaptation has shown that it can be used to infer labels on samples/individuals when no labels are available, suggesting a capacity for violating user privacy. A potential application of UOT is the case where two datasets of users acquired by different methods contain some shared users. UOT can be used here to find correspondences between the users in the two datasets and also identify unique users in each dataset (those that do not receive mass).

## 6   Conclusion

In this paper, we reformulate the UOT problem as a non-negative penalized linear regression, allowing us to propose two new classes of algorithms. We first derive multiplicative algorithms for both KL and $\ell_2$-penalized UOT, providing numerical solutions that are fast and easy to implement. For the specific case of $\ell_2$-penalized UOT, we provide the first regularization path algorithm that computes the whole set of solutions for *all* the regularization parameter values. We finally build on the extensive literature in inverse problem and NMF to draw some fruitful perspectives on even more efficient algorithmic solutions or the definition of new OT problems.

## Acknowledgments and Disclosure of Funding

The authors want to thank Hicham Janati for interesting discussions and providing us with the experiments of convergence for the MM algorithm in the supplemental. This work is partially funded by the French National Research Agency (ANR; grants OATMIL ANR-17-CE23-0012, RAIMO ANR-20-CHIA-0021-01, MULTISCALE ANR-18-CE23-0022-01, E4C ANR-18-EUR-0006-02, 3IA Côte d'Azur ANR-19-P3IA-0002, 3IA ANITI ANR-19-PI3A-0004) and the European Research Council (ERC; grant FACTORY-CoG-6681839). Furthermore, this research was produced within the framework of Energy4Climate Interdisciplinary Center (E4C) of IP Paris and Ecole des Ponts ParisTech. This action benefited from the support of the Chair "Challenging Technology for Responsible Energy" led by l'X - Ecole Polytechnique and the Fondation de l'Ecole Polytechnique, sponsored by TOTAL.

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

# A Supplementary material

## A.1 Design of $H$, $H_r$ and $H_c$

In this section, we detail how we build the design matrix $H$ in problem (3). By setting $\lambda = \lambda_1 = \lambda_2$, Eq. (2) can be reformulated as

$$\text{UOT}^\lambda(\boldsymbol{a}, \boldsymbol{b}) = \min_{\boldsymbol{T} \geq 0} \quad \langle \boldsymbol{C}, \boldsymbol{T} \rangle + \lambda D_\varphi \left( \begin{bmatrix} \boldsymbol{T}\mathbb{1}_m \\ \boldsymbol{T}^\top \mathbb{1}_n \end{bmatrix}, \begin{bmatrix} \boldsymbol{a} \\ \boldsymbol{b} \end{bmatrix} \right) \tag{16}$$

because the divergence $D_\varphi$ is separable. Note that both $\boldsymbol{T}\mathbb{1}_m$ and $\boldsymbol{T}^\top \mathbb{1}_n$ are linear operations. It means that we can vectorize the matrix $\boldsymbol{t} = \text{vec}(\boldsymbol{T}) = [T_{1,1}, T_{1,2}, \ldots T_{n,m-1}, T_{n,m}]^\top$ such that:

$$\begin{bmatrix} \boldsymbol{T}\mathbb{1}_m \\ \boldsymbol{T}^\top \mathbb{1}_n \end{bmatrix} = \boldsymbol{Ht} \qquad \text{where} \qquad \boldsymbol{H} = \begin{bmatrix} \boldsymbol{H}_r \\ \boldsymbol{H}_c \end{bmatrix}. \tag{17}$$

The matrix $\boldsymbol{H}_r \in \mathbb{R}_{n \times nm}$ that performs the sum over the rows of $\boldsymbol{T}$ is given by

$$\boldsymbol{H}_r = \begin{bmatrix} 1 & \ldots & 1 & 0 & \ldots & 0 & \ldots & 0 & \ldots & 0 \\ 0 & \ldots & 0 & 1 & \ldots & 1 & \ldots & 0 & \ldots & 0 \\ \ldots & \ldots & \ldots & \ldots & \ldots & \ldots & \ldots & \ldots & \ldots & \ldots \\ 0 & \ldots & 0 & 0 & \ldots & 0 & \ldots & 1 & \ldots & 1 \end{bmatrix} \tag{18}$$

and can be implemented in Python with $\boldsymbol{H}_r = $ `np.repeat(np.eye(n),m)`. In a similar fashion, the matrix $\boldsymbol{H}_c$ that performs the sum across columns of $\boldsymbol{T}$ is a $m \times nm$ array defined as

$$\boldsymbol{H}_c = [\boldsymbol{I}_m \quad \boldsymbol{I}_m \quad \ldots \quad \boldsymbol{I}_m] \tag{19}$$

and can be implemented in Python using $\boldsymbol{H}_c = $ `np.tile(np.eye(m),n)`.

**Useful identities.** From the previous definitions, we have that

$$\boldsymbol{H}^\top \boldsymbol{y} = \boldsymbol{H}_r^\top \boldsymbol{a} + \boldsymbol{H}_c^\top \boldsymbol{b} = \text{vec}(\boldsymbol{a}\mathbb{1}_m^\top + \mathbb{1}_n \boldsymbol{b}^\top) = \begin{bmatrix} a_1 + b_1 \\ a_1 + b_2 \\ \ldots \\ a_n + b_{m-1} \\ a_n + b_m \end{bmatrix}. \tag{20}$$

We have $\boldsymbol{H}^\top \boldsymbol{H} = \boldsymbol{H}_r^\top \boldsymbol{H}_r + \boldsymbol{H}_c^\top \boldsymbol{H}_c$, of size $nm \times nm$. $\boldsymbol{H}_r^\top \boldsymbol{H}_r$ is a block-diagonal matrix with $n$ blocks of size $m \times m$ filled with ones. $\boldsymbol{H}_r^\top \boldsymbol{H}_r$ can be implemented in Python with `np.tile(np.eye(m),(n,m))`. $\boldsymbol{H}_c^\top \boldsymbol{H}_c$ is a block matrix with blocks of $\boldsymbol{I}_m$, and can be implemented in Python with `np.tile(np.eye(m), (n,n))`. Multiplying $\boldsymbol{H}^\top \boldsymbol{H}$ by a vector, e.g. $\boldsymbol{t}$, results in $\boldsymbol{H}^\top \boldsymbol{Ht} = \text{vec}(\boldsymbol{T}\mathbb{1}_m \mathbb{1}_m^\top + \mathbb{1}_n \mathbb{1}_n^\top \boldsymbol{T})$.

## A.2 Details of MM algorithms

The objective function $F_\lambda(\boldsymbol{t})$ defined by Eq. (3) can be re-written as:

$$F_\lambda(\boldsymbol{t}) = \sum_i \varphi\left(\sum_j H_{i,j} t_j\right) + \sum_j \left[ \frac{c_j}{\lambda} - \sum_i H_{i,j} \varphi'(y_i) \right] t_j. \tag{21}$$

Applying Jensen inequality to the first term like explained in Section 3.1 directly leads to the expression of $G_\lambda(\boldsymbol{t}, \tilde{\boldsymbol{t}})$ given by Eq. (4). The auxiliary function is separable and convex. Given $\tilde{\boldsymbol{t}} = \boldsymbol{t}^{(k)}$, the next iterate $\boldsymbol{t}^{(k+1)}$ can be computed by cancelling the partial derivative $\nabla_{t_q} G_\lambda(\boldsymbol{t}, \boldsymbol{t}^{(k)})$, $q = 1, \ldots, nm$, or setting $t_q$ to zero if the solution is negative in order to satisfy the non-negative constraint (note that this is not a heuristic but what the KKT conditions dictate). Cancelling the partial derivative w.r.t. $t_q$ is equivalent to solving

$$\sum_i H_{i,q} \varphi'\left( \frac{t_q}{t_q^{(k)}} [\boldsymbol{Ht}^{(k)}]_i \right) = \sum_i H_{i,q} \varphi'(y_i) - \frac{c_q}{\lambda} \tag{22}$$

w.r.t. $t_q$. We address this univariate problem for the $\ell_2$ and KL-penalties next.

**Squared $\ell_2$ penalty.** In that case we have $\varphi(x) = \frac{x^2}{2}$, $\varphi'(x) = x$ and we obtain

$$t_q^{(k+1)} = t_q^{(k)} \frac{\max\left(0, [\boldsymbol{H}^\top \boldsymbol{y}]_q - \frac{1}{\lambda} c_q\right)}{[\boldsymbol{H}^\top \boldsymbol{H} \boldsymbol{t}^{(k)}]_q}. \tag{23}$$

Recall that $\boldsymbol{t}$ is a vector form of the OT plan $\boldsymbol{T}$, and assume that $t_q$ corresponds to the entry $T_{i,j}$. $\boldsymbol{H}^\top \boldsymbol{y}$ is a $nm$-dimensional vector with elements $a_i + b_j$, see Eq. (20). Furthermore, we have $\boldsymbol{H}\boldsymbol{t} = \begin{bmatrix} \boldsymbol{T}\mathbb{1}_m \\ \boldsymbol{T}^\top \mathbb{1}_n \end{bmatrix}$ thanks to Eq. (17). Therefore, we can establish the following update in $T_{i,j}$

$$T_{i,j}^{(k+1)} = T_{i,j}^{(k)} \frac{\max\left(0, a_i + b_j - \frac{1}{\lambda} c_{i,j}\right)}{[\boldsymbol{T}^{(k)}\mathbb{1}_m]_i + [\boldsymbol{T}^{(k)\top}\mathbb{1}_n]_j} \tag{24}$$

with matrix form given by Eq. (7).

**KL penalty.** In this case we have $\varphi(x) = x \log x - x$, $\varphi'(x) = \log x$ and we obtain

$$t_q^{(k+1)} = t_q^{(k)} \exp\left(\frac{1}{\sum_q H_{i,q}}\left(\sum_i H_{i,q} \log \frac{y_i}{[\boldsymbol{H}\boldsymbol{t}^{(k)}]_i} - \frac{c_q}{\lambda}\right)\right) \tag{25}$$

$$= t_q^{(k)} \exp\left(\frac{\left[\boldsymbol{H}^\top \log(\boldsymbol{y}) - \boldsymbol{H}^\top \log\left(\boldsymbol{H}\boldsymbol{t}^{(k)}\right)\right]_q - \frac{1}{\lambda} c_q}{\left[\boldsymbol{H}^\top \mathbb{1}\right]_q}\right). \tag{26}$$

Using the results of Section A.1 like in the $\ell_2$ case, we obtain the following update

$$T_{i,j}^{(k+1)} = \left(\frac{a_i}{[\boldsymbol{T}^{(k)}\mathbb{1}_m]_i}\right)^{1/2} T_{i,j}^{(k)} \exp\left(-\frac{c_{i,j}}{2\lambda}\right) \left(\frac{b_j}{[\boldsymbol{T}^{(k)\top}\mathbb{1}_n]_j}\right)^{1/2}$$

with matrix form given by Eq. (6).

**Alternative multiplicative update for the $\ell_2$-penalty.** Another possible approach is to use a quadratic majorization of the linear term $\boldsymbol{c}^\top \boldsymbol{t}$ to bypass the thresholding operation like in (Hoyer, 2002; Yang and Oja, 2011), leading to:

$$\boldsymbol{T}^{(k+1)} = \boldsymbol{T}^{(k)} \odot \frac{\boldsymbol{a}\mathbb{1}_m^\top + \mathbb{1}_n \boldsymbol{b}^\top}{\boldsymbol{T}^{(k)}\boldsymbol{O}_m + \boldsymbol{O}_n \boldsymbol{T}^{(k)} + \frac{1}{2\lambda}\boldsymbol{C}} \quad \text{with} \quad \boldsymbol{O}_\ell = \mathbb{1}_\ell \mathbb{1}_\ell^\top. \tag{27}$$

However we found update (7) more useful in our case, thanks to the thresholding operation that locates true zeros from start.

**Alternative derivation of MM algorithms.** The reformulation of UOT as a non-negative penalized linear regression problem comes very handy because it offers a novel interpretation of UOT and the possibility of using some of the many existing algorithms for the latter problem, such as LARS-based algorithm for path computation. However, we want to point out that we may also derive MM algorithms directly from Eq. (2). Let us write

$$F_\lambda(\boldsymbol{T}) = \langle \boldsymbol{C}, \boldsymbol{T} \rangle + \lambda_1 D_\varphi(\boldsymbol{T}\mathbb{1}_m, \boldsymbol{a}) + \lambda_2 D_\varphi(\boldsymbol{T}^\top \mathbb{1}_n, \boldsymbol{b}) \tag{28}$$

$$= \sum_{ij} C_{i,j} T_{i,j} + \lambda_1 \sum_i d_\varphi(\sum_j T_{i,j}, a_i) + \lambda_2 \sum_j d_\varphi(\sum_i T_{i,j}, b_j) \tag{29}$$

(Note that we have $F_\lambda(\boldsymbol{T}) = F_\lambda(\boldsymbol{t})$, slightly abusing notations). Let $\tilde{\boldsymbol{T}}$ be a current estimate of $\boldsymbol{T}$. We wish to compute an auxiliary function $G_\lambda(\boldsymbol{T}, \tilde{\boldsymbol{T}})$ for $F_\lambda(\boldsymbol{T})$. Let us denote

$$\tilde{a}_i = \sum_j \tilde{T}_{i,j} \quad \text{(the } i^{th} \text{ approximate row marginal)} \tag{30}$$

$$\tilde{b}_j = \sum_i \tilde{T}_{i,j} \quad \text{(the } j^{th} \text{ approximate column marginal)} \tag{31}$$

$$\tilde{\alpha}_{i,j} = \frac{\tilde{T}_{i,j}}{\tilde{a}_i} \qquad \text{such that } \sum_j \tilde{\alpha}_{i,j} = 1 \tag{32}$$

$$\tilde{\beta}_{i,j} = \frac{\tilde{T}_{i,j}}{\tilde{b}_j} \qquad \text{such that } \sum_i \tilde{\beta}_{i,j} = 1 \tag{33}$$

By convexity of $d_\varphi(x, y)$ w.r.t $x$, we have

$$d_\varphi \left( \sum_j T_{i,j}, a_i \right) \leq \sum_j \tilde{\alpha}_{i,j} \, d_\varphi \left( \frac{T_{i,j}}{\tilde{\alpha}_{i,j}}, a_i \right), \tag{34}$$

$$d_\varphi \left( \sum_i T_{i,j}, b_j \right) \leq \sum_i \tilde{\beta}_{i,j} \, d_\varphi \left( \frac{T_{i,j}}{\tilde{\beta}_{i,j}}, b_j \right). \tag{35}$$

The inequalities are tight when $\tilde{\boldsymbol{T}} = \boldsymbol{T}$. Plugging the latter inequalities into Eq. (29), we obtain the following auxiliary function:

$$G_\lambda(\boldsymbol{T}|\tilde{\boldsymbol{T}}) = \sum_{ij} \left[ C_{i,j} T_{i,j} + \lambda_1 \tilde{\alpha}_{i,j} \, d_\varphi \left( \frac{T_{i,j}}{\tilde{\alpha}_{i,j}}, a_i \right) + \lambda_2 \tilde{\beta}_{i,j} \, d_\varphi \left( \frac{T_{i,j}}{\tilde{\beta}_{i,j}}, b_j \right) \right]. \tag{36}$$

$G_\lambda(\boldsymbol{T}|\tilde{\boldsymbol{T}})$ is essentially the matrix form of $G_\lambda(\boldsymbol{t}|\tilde{\boldsymbol{t}})$, with partial derivative given by:

$$\nabla_{T_{i,j}} G_\lambda(\boldsymbol{T}|\tilde{\boldsymbol{T}}) = C_{i,j} + \lambda_1 d'_\varphi \left( \tilde{a}_i \frac{T_{i,j}}{\tilde{T}_{i,j}}, a_i \right) + \lambda_2 d'_\varphi \left( \tilde{b}_j \frac{T_{i,j}}{\tilde{T}_{i,j}}, b_j \right). \tag{37}$$

Using $d'_\varphi(x, y) = \varphi'(x) - \varphi'(y)$ and either $\varphi'(x) = x$ ($\ell_2$-penalized UOT) or $\varphi'(x) = \log x$ (KL-penalized UOT), we easily retrieve Eq. (24) and Eq. (27) when $\lambda_1 = \lambda_2$, or Eq. (14) in the general case (with here $\lambda_{\text{reg}} = 0$).

### A.3 Details of the UOT path computation

**Matrices and vectors on the active set $\mathcal{A}$.** Recall that $\boldsymbol{m}_\mathcal{A}$, $\boldsymbol{c}_\mathcal{A}$ and $\boldsymbol{t}_\mathcal{A}$ are sub-vectors of $\boldsymbol{m}$, $\boldsymbol{c}$ and $\boldsymbol{T}$ corresponding to indices in $\mathcal{A}$. $\boldsymbol{H}_\mathcal{A}$ is a matrix of dimension $(|i| + |j|) \times |\mathcal{A}|$, where $|i|$ and $|j|$ are respectively the number of distinct rows $i$ and columns $j$ that belong to the transport plan for a given active set $\mathcal{A}$. $\boldsymbol{H}_\mathcal{A}$ is built by keeping only the rows of $\boldsymbol{H}_r$ such that the element $i$ is present in the active set (the latter matrix being denoted $[\boldsymbol{H}_r]_\mathcal{A}$), the rows of $\boldsymbol{H}_c$ such that the element $j$ is present in the active set (denoted $[\boldsymbol{H}_c]_\mathcal{A}$), and keeping the columns such that element $(i, j) \in \mathcal{A}$ (up to vectorization).

**Update $(\boldsymbol{H}_\mathcal{A}^\top \boldsymbol{H}_\mathcal{A})^{-1}$ from $(\boldsymbol{H}_{\mathcal{A}_k}^\top \boldsymbol{H}_{\mathcal{A}_k})^{-1}$ using the Schur complement.** Algorithm 1 involves the computation, at each iteration, of the inverse matrix $(\boldsymbol{H}_\mathcal{A}^\top \boldsymbol{H}_\mathcal{A})^{-1}$. The computational burden can be alleviated by using the Schur complement of the matrix in order to compute $(\boldsymbol{H}_\mathcal{A}^\top \boldsymbol{H}_\mathcal{A})^{-1}$ from its value at the previous iteration $(\boldsymbol{H}_{\mathcal{A}_k}^\top \boldsymbol{H}_{\mathcal{A}_k})^{-1}$. Let us denote $\boldsymbol{B}_\mathcal{A} = (\boldsymbol{H}_\mathcal{A}^\top \boldsymbol{H}_\mathcal{A})$ and $\boldsymbol{B}_{\mathcal{A}_k} = (\boldsymbol{H}_{\mathcal{A}_k}^\top \boldsymbol{H}_{\mathcal{A}_k})$. Two cases may arise:

- One component $q$ is added to the active set $\mathcal{A} = \mathcal{A}_{k+1} = \mathcal{A}_k \cup q$. In that case, we have:

$$\boldsymbol{B}_\mathcal{A}^{-1} = \begin{bmatrix} \boldsymbol{B}_{\mathcal{A}_k}^{-1} + \boldsymbol{B}_{\mathcal{A}_k}^{-1} b_{\mathcal{A},q} S^{-1} b_{q,\mathcal{A}} \boldsymbol{B}_{\mathcal{A}_k}^{-1} & -\boldsymbol{B}_{\mathcal{A}_k}^{-1} b_{\mathcal{A},q} S^{-1} \\ -S^{-1} b_{q,\mathcal{A}} \boldsymbol{B}_\mathcal{A}^{-1} & S^{-1} \end{bmatrix} \tag{38}$$

where $b_{q,\mathcal{A}}$ is the last column of matrix $\boldsymbol{B}_\mathcal{A}$, $b_{\mathcal{A},q}$ its last row and $S = 2 - b_{q,\mathcal{A}}{}^\top \boldsymbol{B}_{\mathcal{A}_k}^{-1} b_{\mathcal{A},q}$ is a scalar.

- One component $q$ is removed from the active set $\mathcal{A} = \mathcal{A}_k \backslash q$. In that case, we get:

$$\boldsymbol{B}_{\mathcal{A}}^{-1} = \boldsymbol{B}_{\mathcal{A}_k \backslash q}^{-1} - \frac{b_{\mathcal{A}\backslash q, q}^{-1} b_{q, \mathcal{A}\backslash q}^{-1}}{b_{q,q}^{-1}} \tag{39}$$

with $\boldsymbol{B}_{\mathcal{A}\backslash q}^{-1}$ being the matrix $\boldsymbol{B}_{\mathcal{A}}^{-1}$ deprived from its row and column corresponding to the component $q$. The vector $b_{\mathcal{A}_k \backslash q, q}^{-1}$ represents the column of the $\boldsymbol{B}_{\mathcal{A}}^{-1}$ matrix corresponding to element $i$ while $b_{q, \mathcal{A}\backslash q}^{-1}$ stands for the corresponding row. Finally $b_{q,q}^{-1}$ is the component of $\boldsymbol{B}_{\mathcal{A}}^{-1}$ corresponding to the component $q$.

## A.4 Details of the regularization path formulation for semi-relaxed UOT

**Semi-relaxed $\ell_2$-penalized UOT.** We start by recalling the formulation of the semi-relaxed $\ell_2$-penalized UOT problem:

$$\text{SROT}^{\boldsymbol{\lambda}}(\boldsymbol{a}, \boldsymbol{b}) = \min_{\boldsymbol{T} \geq 0, \boldsymbol{H}_c \boldsymbol{t} = \boldsymbol{b}} \langle \boldsymbol{C}, \boldsymbol{T} \rangle + \lambda \|\boldsymbol{T}\mathbb{1}_m - \boldsymbol{a}\|^2.$$

From Eq. (12), the corresponding Lagrangian writes:

$$L_\lambda(\boldsymbol{t}, \boldsymbol{\gamma}) = \frac{1}{\lambda}\boldsymbol{c}^\top \boldsymbol{t} + \frac{1}{2}(\boldsymbol{H}_r \boldsymbol{t} - \boldsymbol{a})^\top (\boldsymbol{H}_r \boldsymbol{t} - \boldsymbol{a}) + (\boldsymbol{H}_c \boldsymbol{t} - \boldsymbol{b})^\top \boldsymbol{u} - \boldsymbol{\gamma}^\top \boldsymbol{t} \tag{40}$$

with $\boldsymbol{u} \in \mathbb{R}^m$ the Lagrange parameters associated to the $m$ equality constraints and $\boldsymbol{\gamma} \geq 0$ the Lagrange parameters related to the non-negativity constraints. We recall the KKT optimality conditions, which state that i) $\nabla_t L_\lambda = \frac{1}{\lambda}\boldsymbol{c} + \boldsymbol{H}_r^\top \boldsymbol{H}_r \boldsymbol{t} - \boldsymbol{H}_r^\top \boldsymbol{a} + \boldsymbol{H}_c^\top \boldsymbol{u} - \boldsymbol{\gamma} = 0$ (stationary condition), ii) $\boldsymbol{\gamma} \odot \boldsymbol{t} = 0$ (complementary condition), and iii) $\boldsymbol{\gamma} \geq 0$ and $\boldsymbol{H}_c \boldsymbol{t} - \boldsymbol{b} = \boldsymbol{0}$ (feasibility) from which we may derive the path computation. We recall that $\odot$ stands for point-wise multiplication.

**Piecewise linearity of the path.** Let us suppose that, at step $k$, we know the current active set $\mathcal{A} = \mathcal{A}_k$ and we look for $\boldsymbol{t}_{\mathcal{A}}^\lambda$ and $\boldsymbol{u}^\lambda$. Because of the complementary condition, we have $\boldsymbol{\gamma}_{\mathcal{A}} = \boldsymbol{0}$. Hence the stationnarity condition on the active set can be rewritten as, with $\lambda = \lambda_k + \epsilon$ and $\epsilon$ small enough

$$\begin{cases} [\boldsymbol{H}_r^\top]_{\mathcal{A}} [\boldsymbol{H}_r]_{\mathcal{A}} \boldsymbol{t}_{\mathcal{A}}^\lambda + [\boldsymbol{H}_c^\top]_{\mathcal{A}} \boldsymbol{u}^\lambda &= [\boldsymbol{H}_r^\top]_{\mathcal{A}} \boldsymbol{a}_{\mathcal{A}} - \frac{1}{\lambda}\boldsymbol{c}_{\mathcal{A}} \\ [\boldsymbol{H}_c]_{\mathcal{A}} \boldsymbol{t}_{\mathcal{A}}^\lambda &= \boldsymbol{b}_{\mathcal{A}} \end{cases} \tag{41}$$

or equivalently, at each iteration, the following linear system should be solved:

$$\underbrace{\begin{pmatrix} [\boldsymbol{H}_r^\top]_{\mathcal{A}} [\boldsymbol{H}_r]_{\mathcal{A}} & [\boldsymbol{H}_c^\top]_{\mathcal{A}} \\ [\boldsymbol{H}_c]_{\mathcal{A}} & \boldsymbol{0} \end{pmatrix}}_{\boldsymbol{K}_{\mathcal{A}}} \begin{pmatrix} \boldsymbol{t}_{\mathcal{A}}^\lambda \\ \boldsymbol{u}^\lambda \end{pmatrix} = -\frac{1}{\lambda} \underbrace{\begin{pmatrix} \boldsymbol{c}_{\mathcal{A}} \\ \boldsymbol{0} \end{pmatrix}}_{\boldsymbol{\gamma}_{\mathcal{A}}} + \underbrace{\begin{pmatrix} [\boldsymbol{H}_r^\top]_{\mathcal{A}} \boldsymbol{a}_{\mathcal{A}} \\ \boldsymbol{b} \end{pmatrix}}_{\boldsymbol{\beta}_{\mathcal{A}}}. \tag{42}$$

We then have

$$\begin{pmatrix} \boldsymbol{t}_{\mathcal{A}}^\lambda \\ \boldsymbol{u}^\lambda \end{pmatrix} = -\frac{1}{\lambda} \boldsymbol{K}_{\mathcal{A}}^{-1} \boldsymbol{\gamma}_{\mathcal{A}} + \boldsymbol{K}_{\mathcal{A}}^{-1} \boldsymbol{\beta}_{\mathcal{A}}. \tag{43}$$

We now denote $\tilde{\boldsymbol{c}}_{\mathcal{A}} = \boldsymbol{K}_{\mathcal{A}}^{-1} \boldsymbol{\gamma}_{\mathcal{A}}$ and its sub-vectors $\tilde{\boldsymbol{c}}_{\mathcal{A}}^a$ and $\tilde{\boldsymbol{c}}_{\mathcal{A}}^b$ that respectively contains the $|\mathcal{A}|$ first rows and $m$ last rows of $\tilde{\boldsymbol{c}}_{\mathcal{A}}$. We also denote $\tilde{\boldsymbol{m}}_{\mathcal{A}} = \boldsymbol{K}_{\mathcal{A}}^{-1} \boldsymbol{\beta}_{\mathcal{A}}$ and its sub-vectors $\tilde{\boldsymbol{m}}_{\mathcal{A}}^a$ and $\tilde{\boldsymbol{m}}_{\mathcal{A}}^b$ in the same fashion. We then have

$$\begin{cases} \boldsymbol{t}_{\mathcal{A}}^\lambda = -\frac{1}{\lambda}\tilde{\boldsymbol{c}}_{\mathcal{A}}^a + \tilde{\boldsymbol{m}}_{\mathcal{A}}^a \\ \boldsymbol{u}^\lambda = -\frac{1}{\lambda}\tilde{\boldsymbol{c}}_{\mathcal{A}}^b + \tilde{\boldsymbol{m}}_{\mathcal{A}}^b \end{cases} \tag{44}$$

We again notice the piecewise linearity (as a function of $1/\lambda$) of the path when the active set $\mathcal{A}$ is fixed.

**Computation of $\lambda^{k+1}$ given $\lambda^k$.** Given a current solution at iteration $k$ $(\lambda_k, t^{\lambda_k})$, we increase the $\epsilon$ value in $\lambda = \lambda_k + \epsilon$ until one of the following case arises.

• **Inside the active set,** the positivity constraint on $t_{\mathcal{A}}^{\lambda}$ may be violated, corresponding to the case

$$\tilde{m}_{\mathcal{A}}^a = \frac{1}{\lambda}\tilde{c}_{\mathcal{A}}^a \quad \Rightarrow \quad \lambda_r = \min_{>\lambda_k} \left( \frac{\tilde{c}_{\mathcal{A}}^a}{\tilde{m}_{\mathcal{A}}^a} \right) \tag{45}$$

where $\min_{>\lambda_k}$ denotes the smallest value in $\frac{\tilde{c}_{\mathcal{A}}^a}{\tilde{m}_{\mathcal{A}}^a}$ greater that $\lambda_k$.

• **Outside the active set,** the positivity constraint of the KKT may be violated. The stationarity condition outside the active set $\bar{\mathcal{A}}$ can be rewritten, by injecting the solution of Eq. (44):

$$\frac{1}{\lambda}c_{\bar{\mathcal{A}}} + \left[ H_r^\top (H_r(-\frac{1}{\lambda}\tilde{c}^a + \tilde{m}^a) - a) \right]_{\bar{\mathcal{A}}} + \left[ H_c^\top (-\frac{1}{\lambda}\tilde{c}^b + \tilde{m}^b) \right]_{\bar{\mathcal{A}}} - \gamma_{\bar{\mathcal{A}}} = 0 \tag{46}$$

$$\frac{1}{\lambda}c_{\bar{\mathcal{A}}} + \left[ H^\top H (\tilde{m} + \frac{1}{\lambda}\tilde{c}) \right]_{\bar{\mathcal{A}}} - m_{\bar{\mathcal{A}}} = \gamma_{\bar{\mathcal{A}}} \quad \Rightarrow \quad \lambda_a = \min_{>\lambda_k} \left( \frac{c_{\bar{\mathcal{A}}} - [H^\top H \tilde{c}]_{\bar{\mathcal{A}}}}{m_{\bar{\mathcal{A}}} - [H^\top H \tilde{m}]_{\bar{\mathcal{A}}}} \right) \tag{47}$$

The active set changes only if there exists a component $i$ outside the current active set such that $\gamma_i \geq 0$. Hence we write:

$$\frac{1}{\lambda}c_{\bar{\mathcal{A}}} - \frac{1}{\lambda}\left[ H_r^\top H_r \tilde{c}^a + H_c^\top \tilde{c}^b \right]_{\bar{\mathcal{A}}} + \left[ H_r^\top H_r \tilde{m}^a - H_r^\top a + H_c^\top \tilde{m}^b \right]_{\bar{\mathcal{A}}} \geq 0 \tag{48}$$

$$\lambda_a = \min_{>\lambda_k} \frac{c_{\bar{\mathcal{A}}} - \left[ H_r^\top H_r \tilde{c}^a + H_c^\top \tilde{c}^b \right]_{\bar{\mathcal{A}}}}{\left[ 2H_r^\top a - H_r^\top H_r \tilde{m}^a - H_c^\top \tilde{m}^b \right]_{\bar{\mathcal{A}}}} \tag{49}$$

Note that this last equation is very similar to the one we obtain for $\ell_2$-penalized UOT, except that vectors $\tilde{m}$ and $\tilde{c}$ are split in 2 parts, depending on if we consider the rows (that can be unbalanced) or the columns (that should strictly respect the marginal constraint). Also note that the Schur complement applies to the update of $K_{\mathcal{A}}^{-1}$ in order to decrease the computational burden.

## A.5 IPOT is a MM algorithm

Herein we discuss the relation between the Inexact Proximal Point OT (IPOT) algorithm of Xie et al. (2020) and MM. First note that IPOT aims at the balanced OT problem (1). This is equivalent to solving

$$\min_{T \geq 0, T\mathbb{1}_m = a, T^\top \mathbb{1}_n = b} \langle C, T \rangle \quad + \lambda \sum_{i,j} T_{i,j} \log(T_{i,j}) - \lambda \sum_{i,j} T_{i,j} \log(T_{i,j}) \tag{50}$$

where one adds and removes the entropy regularization of $T$. A simple algorithm can be devised by upper bounding the concave term by its tangent at $T^{(k)}$ leading to the new problem

$$\min_{T \geq 0, T\mathbb{1}_m = a, T^\top \mathbb{1}_n = b} \langle T, C \rangle + \lambda \sum_{i,j} T_{i,j} \log(T_{i,j}) - \lambda \langle T, \log\left( T^{(k)} \right) + 1 \rangle \tag{51}$$

where the log is taken component-wise. Note that the constant 1 in the scalar product can be removed since $\sum_{i,j} T_{i,j}$ is constant and does not influence the solution. Problem (51) can be solved using classical Sinkhorn iterations with a cost matrix $\tilde{C} = C - \lambda \log(T^{(k)})$. This corresponds to using the kernel matrix

$$\tilde{K} = \exp\left( -\frac{1}{\lambda}(C - \lambda \log\left( T^{(k)} \right)) \right) = \exp\left( -\frac{1}{\lambda}C \right) \odot T^{(k)}, \tag{52}$$

as presented in (Xie et al., 2020, Algorithm 1). Hence IPOT can be interpreted as MM. Note that the point-wise product between a kernel matrix and the estimate $T^{(k)}$ appears also in our multiplicative updates (6) and (14) with however a different scaling parameter.

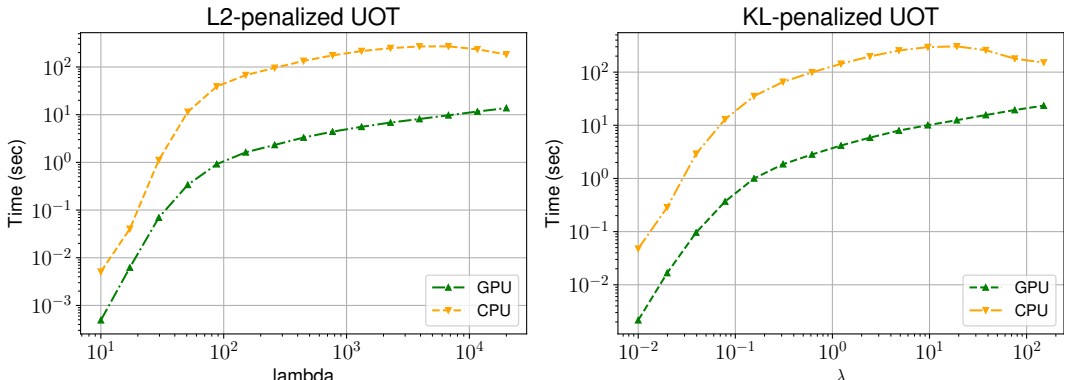

Figure 5: (Left) Comparison of $\ell_2$-penalized UOT with $m = n = 500$ run on CPU and GPU (Right) likewise for KL-penalized UOT. Values represent average values over 5 runs.

### A.6 Details about the experiments in the paper

We run the experiments on a Mac mini 2020 personal computer, with M1 chip and 16GB of RAM. All the experiments can be re-run thanks to the paper companion code. We compare the following algorithms provided by the following solvers:

- the "L-BFGS-B" method of SciPy, in which we provide the function to minimize and its associated Jacobian (either for KL or $\ell_2$-penalized UOT),
- the Lasso algorithms Celer and of Scikit-learn,
- the regularization path algorithm introduced in the paper,
- the multiplicative updates introduced in the paper.

We use the same stopping criteria for all the algorithms (not to mention the regularization path algorithm that provides an exact solution), except for $\ell_2$-penalized UOT that necessitates a smaller tolerance to converge to the correct values, especially for large values of $\lambda$.

Regarding Figure 3, we draw 5 realizations of two random 2 Gaussian samples of 10-dimensional $n = m$ points with different means and variances.

### A.7 Additional experiments

**GPU implementation of the MM algorithms.** Figure 5 compares the results obtained by running the MM algorithms on CPU and GPU (GeForce GTX TITAN X), showing that it is about 5 to 40 times faster to run the MM UOT algorithm on GPU.

**Convergence of the MM algorithm to a closed form solution.** As discussed in the introduction, there exist closed form solutions for KL-UOT betwen Gaussians for the regularized Janati et al. (2020) and unregularized (Janati, 2021, Eq. 2.72) UOT. The second one for unregularized UOT allows us to use the closed form solution to check that we converge to the true UOT value when the number of samples $n$ goes to infinity. To this end, we simulate 50 realizations of samples drawn from Gaussian distributions in dimension $d = 1, 2, 4$ and study the evolution of the error of the OT loss as a function of the number of samples $n$ for different values of the regularization parameter $\lambda$. For the Gaussian distribution, we take a mean $\boldsymbol{\mu}$ equal to a null vector and we draw the covariance $\boldsymbol{\Sigma}$ from a Wishart distribution. Note that the error for all configurations decreases, suggesting that our algorithm can recover the true UOT value.

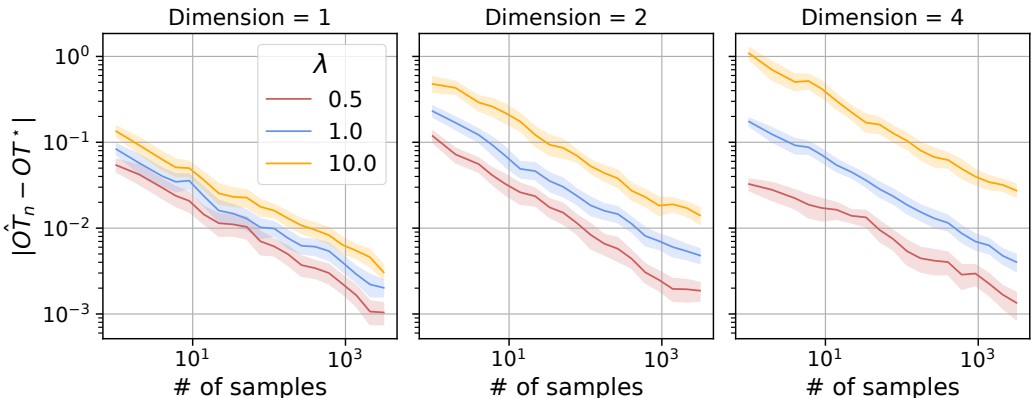

Figure 6: Illustration of the empirical convergence of the KL UOT to its continuous closed form solution between Gaussians using our KL-UOT MM solver. Absolute errors (and related variance) are provided for different realizations with data dimensionality equal to 1 (left), 2 (center) and 4 (right) as a function of the number of samples $n$.