# OpenReview forum: "Unbalanced Optimal Transport through Non-negative Penalized Linear Regression"
_NeurIPS.cc/2021/Conference — NeurIPS 2021 Poster_

### Official Review · Reviewer_7tpv · 2021-07-07

**Rating:** 6
**Confidence:** 3

**Summary:**

In this work, the authors propose to phrase the Unbalanced Optimal Transport problem under general Bregman divergence penalty as a non-negative penalized linear regression problem. They exploit an existing algorithm in Non-negative Matrix Factorization problem (NMF), namely Majorization-Minimization, in order to solve the obtained problem. In the case of KL-penalized UOT, the obtained algorithm is a variation of the Sinkhorn algorithm. However, this algorithm solves a different algorithm than Sinkhorn, namely KL-penalized UOT as opposed to OT with KL penalized transport plan.  For the l2-penalized UOT, the authors design an algorithm, based on LARS, that solves the problem for all values of lambda, from 0 to infinity.

The running time of different variants of their proposed algorithm are compared on a toy problem versus the current state-of-the-art method that uses BFGS. Overall, for l2-penalized UOT, the proposed method is significantly faster than BFGS. For KL-penalized UOT, it depends on the value of lambda.


**Limitations And Societal Impact:**

The authors adequately addressed the limitations and potential negative societal impact.

**Main Review:**

pros:
- The connection made between UOT and non-negative linear regression allows for the use of well-studied efficient algorithms.
For KL-penalized UOT, the algorithm solves the problem without needing to regularize the transport plan, as opposed to using Sinkhorn divergence for UOT as done in [1].
- For l2-regularized UOT, the proposed algorithm can be solved once for all values of the regularization parameter, and is shown to be faster than the state-of-the-art method that uses BFGS.

cons:
- The improved performance of the algorithm is shown only in the case of l2-penalized UOT, whose practical interest is not clear compared to KL-penalized UOT, which is more commonly seen in the literature.
- In the case of KL-penalized UOT, I expected a deeper analysis of the connection with the Sinkhorn algorithm and the Sinkhorn divergence based UOT of [1], e.g., how similar do they become for asymptotic values of lambda and epsilon (epsilon being the reg. parameter for the entropy of the transport plan).

Typos:

l.319 balanced OT

[1] Sinkhorn Divergences for Unbalanced Optimal Transport, Séjourné et al.


**Time Spent Reviewing:**

3h

---

> ### Author Response · Authors · 2021-08-10
> **Answer to reviewer 7tpv**
>
> ### Improvement only shows in $\ell_2$-penalized UOT, but unclear in $KL$-penalized UOT
>
> According to Fig 3, right panel, we have a very important gain with MM in the small $\lambda $ regime that can be of interest in practice (to detect out of distribution samples for instance). In this case LBFGS is slow because the problem is less and less quadratic/non linear and the quasi newton approximation makes less and less sense.
>
> We also think that the MM algorithm leads to very simple updates that can be implemented with just a few lines of code and use only matrix and pointwise products that have been heavily optimized on modern hardware (CPU and GPU) whereas more involved algorithms such as LBFGS will require more complex and hard to differentiate steps such as line search operations. (same answer to reviewer 9n5u)
>
>
> ### A deeper comparaison with the Sinkhorn algorithm for KL-UOT (Séjourné et al.)
>
> It is true that we compare the performances of our algorithms only to solvers that provide the *exact unbalanced transport plans* as it is an important feature sought in some applications (PU learning as in Chapel et al., robust optimal transport as in Mukherjee et al.). Sejourné et al. deal with entropic-regularized unbalanced optimal transport, that allows constructing fast solvers. But when the entropic regularisation parameter becomes small and converge to $0$ (providing a sparser transport plan), the algorithm becomes a fixed point that cannot converge to a solution of the porblem. In other words the algorithm from Sejourné et al. cannot be used when $\lambda_{reg}\rightarrow 0$ but our alternative update proposed in equation (14) converges continuously to our proposed MM iteration Eq. (6) in this case. We will add a discussion in the paper.
>
> Chapel, l., Alaya, M., Gasso, G. Partial optimal tranport with application to Positive-Unlabeled learning. NeurIPS 2020

---

### Official Review · Reviewer_42ef · 2021-07-15

**Rating:** 6
**Confidence:** 3

**Summary:**

This paper proposes a new method for Unbalanced Optimal Transport (UOT) without a regularization term. The solution consists of two steps; (i) formulate the UOT as a non-negative penalized linear regression problem (ii) solve it using the Majorization-Minimization (MM) algorithm. The authors derive specific algorithms for several penalty functions. Furthermore, a regularization path tracking method based on the relationship between the proposed formulation and lasso is proposed. The authors validate the effectiveness of the proposed method by numerical experiments using synthetic datasets and the MNIST dataset.

**Limitations And Societal Impact:**

The limitation of this paper is that it has little advantage over existing methods in terms of computation time, as the authors also state in the checklist.  The societal impacts are fully discussed at the end of Section 5.

**Main Review:**

### Strengths

* The idea of formulating UOT as a non-negative penalized linear regression problem and applying the MM algorithm is very interesting.

* Also, there is originality in the regularized path tracking algorithm based on the above formulation.

* This paper is well-written; the problems, the backgrounds, and the contributions are clearly stated with appropriate citations.

### Weaknesses

* The analysis of the time complexity is insufficient both theoretically and experimentally. The theoretical analysis of the time complexity for each iteration and convergence is not sufficiently done for any of the methods, and the superiority of the proposed method cannot be judged. As for the experiments, the scalability of the proposed method cannot be judged from the experimental results because the comparison with the existing methods is made only when \lambda is changed, and there is no comparison when n or m is changed. Although the authors do not claim that the computation time is an advantage of the proposed method, these analyses are necessary to understand the characteristics of the proposed method.

* In the introduction, the authors state that one of the advantages of the proposed method is fast computation by GPUs, but this claim has not been verified theoretically or experimentally.

* In the introduction, the ease of implementation is mentioned as an advantage of the proposed method, but since the L-BFGS method can also be easily implemented by using scipy, there seems to be no significant difference.

### Other comments

* Regularized path in OT is interesting, but it is a bit unclear how it can be useful. I would like to know if there are any specific use cases.

* In the experiments, the authors compare their method with the BFGS method in Blondel et al. (2018).  Blondel et al. (2018) propose two methods in the paper, one for solving the dual and one for solving the semi-dual.  Which method is used in the experiment? In addition, Blondel et al. (2018) also describe an optimization method using alternating minimization. Because this method resembles the proposed method, shouldn't it be compared with this method as well?

* Section 5 contains various speed-up proposals for the proposed algorithm.  A thorough theoretical and experimental evaluation of these proposals will make this paper more interesting.

**Time Spent Reviewing:**

7

---

> ### Author Response · Authors · 2021-08-10
> **Answer to Reviewer 42ef**
>
> ### Insufficient analysis of theoretical complexity and convergence
> The theoretical study of MM algorithms have been presented in the literature (see Dhillon and S. Sra (2005), Sun et al. (2017), Zhao and Tan (2018)). Under mild-conditions satisfied by our studied UOT, MM algorithms  are  guaranteed  to converge.
> For the regularization path of OT, a complexity analysis has been done in the work of Mairal and Yu (2012). In the latter reference it is established that in the worst case  scenario  a combinatorial but finite number of iterations may be required to compute the entire solution path. As discussed in Section 5, an acceleration can be achieved by computing an approximate regularization path with $\epsilon$-precision in $O(1/\epsilon)$ iterations.
> We will make these points clearer in the final version.
>
> Sun, Ying, Prabhu Babu, and Daniel P. Palomar. "Majorization-minimization algorithms in signal processing, communications, and machine learning." IEEE Transactions on Signal Processing 65.3 (2016): 794-816
>
> R. Zhao and V. Y. F. Tan, "A Unified Convergence Analysis of the Multiplicative Update Algorithm for Regularized Nonnegative Matrix Factorization," in IEEE Transactions on Signal Processing, vol. 66, no. 1, pp. 129-138, 1 Jan.1, 2018, doi: 10.1109/TSP.2017.2757914.
>
>
> ### Insufficient analysis of the experimental complexity
>
> OT problems  are notoriously known to be computationaly intensive unless specific regularizations (such as entropy regularization) are used to approximate the original OT problem. The UOT may suffer a  similar issue.  Based on that, we intend in the paper to provide some alternatives to solve the UOT problem in the general case (even non-regularized). The empirical analysis shows the benefits of the proposed optimization methods  w.r.t. the $\lambda$-regime (see  Figure 3). Notice that the derived multiplicative updates render our MM-method amenable to GPU  computation. We will add a figure illustrating complexity as a functionn of $n$ but note that our preliminary experiments suggets that all methods have a similar complxity w.r.t. $n$.
>
> ### Lack of implementation with GPUs
>  We agree that  such  implementation is not provided in  the paper.  Nevertheless, the  proposed multiplicative updates are amenable to  parallel computation similarly to Sinkhorn-Knopp updates  or  L-BFGS method. We believe that the provided CPU evaluations reflect the behavior of compared algorithms when lifted to GPU computation (where the complexity of LBFGS due to linsearch can be a limiting factor). Note that the paper introduces a novel framework and evluates several new alternatives to solve UOT. We decided to leave speed ups and implementation improvements (such as GPU implementation) for future works.
>
> Also we feel that requiring a GPU implementation in a paper is unfair as GPU access is source of inequality in research. Also such requirement would force reseachers with no GPU access to run their private code and data on free access GPUs such as Google Colab.
>
>
> ### Ease of implementation comparing with L-BFGS method
> We also think that the MM algorithm leads to very simple updates that can be implemented with just a few lines of code and use only matrix and pointwise products that have been heavily optimized on modern hardware (CPU and GPU) whereas more involved algorithms such as LBFGS will require more complex and hard to differentiate steps such as line search operations.
>
> Although scipy facilitate the implementation of the much more complex L-BFGS,  we can see, from Figure 3 middle and right, L-BFGS is clearly not the best choice of solver in practice in the majority of configurations.
>
>
> ### Other comments
>
> - Use of regularization path of OT
>
> Having a continuity of solutions for the whole regularization path is a tremendous result in machine learning (ML) as the hyper-parameters of all ML methods have to be validated in practice which require a lot of computation power. When looking at the center panel of Figure 3 we can see that solving the UOT for large $\lambda $ is is slightly faster with Lasso solvers (which is also a contribution since no research work formulates the UOT with quadratic divergence as a Lasso), but those solvers only provide a solution for a unique value of $\lambda $. Estimating the relevant $\lambda $ given a UOT task might be very costly whereas the regularization path provides the whole solution path for a fairly limited numerical cost. Finally, searching for “regularization path” in Google scholar returns more than 4000 papers which illustrates the interest of the approach in practice and our paper is the first to provide a regularization path algorithm to solve OT problem to the best of our knowledge.
>
> - Which formulation of Blondel et al. is solved?
>
>
> The L-BFGS method is used to solve the relaxed smooth primal problem (Def.2 in Blondel et al.,2018). None of the dual formulations is involved in our experiments.
>
> - Compare our method with alternating minimization method described in Blondel et al.,2018?
>
> The alternating minimization with block update is interesting in this case but it requires the used of a simplex projection that is highly not differentiable (smooth versions might exist but is out of the scopre of the paper). Our operations in the MM algorithm are much simpler, differentiable for KL and non-diffeerntiable but rely on a ReLu for L2 wrt the input parameter (see Eq. 7) that are known to work well in practice in deep neural networks.
>
> - add experimental and theoretical evaluation for the speed-up proposals in section 5
>
> We strongly agree that those speed up proposals are of strong interest. Nevertheless, we aim in this paper  at providing a new framework and new algorithms to solve the UOT problem and at evaluating their complexities w.r.t. standard algorithms that solve the same problem. As it is, we believe that the paper answer these points. We then leave speed ups and implementation improvements (such as GPU implementation) for future works since such endeavours will clearly not fit the page requirement of Neurips.

---

> > ### Comment · Reviewer_42ef · 2021-08-27
> > **I am willing to raise my score**
> >
> > I have read the other reviewers' reviews and the authors' rebuttal.
> >
> > Many of my concerns have been addressed by the rebuttal.
> >
> > Provided that the following information which the authors explain in their rebuttal is added, I am willing to raise the score to 6.
> >
> > * Results of the theoretical analysis of the MM algorithm
> > * Additional experimental results on computation time, especially for the case varying n and m.
> > * Practical motivation for regularization path
> >
> > In addition, although it may not be fair to force GPU experiments, we believe that GPU experiments are necessary to reinforce the claims of this paper and to make it a better paper. If possible, the final version of the paper should include the results of the GPU experiments.

---

> > > ### Author Response · Authors · 2021-08-27
> > > **Some results for GPU implementation**
> > >
> > > Thanks for your feedback. We will add the requested discussions and experiments (running times for different values of n,m, emphasize the advantages of the MM algorithm, more discussion about the interest of the regularization path).
> > > As advised, we did some preliminary experiments for the MM algorithm for solving KL-UOT on GPU (using Google colab). Here are the results averaged on 5 runs, keeping the same setting as Fig. 3, right panel:
> > >
> > > |    $\lambda$ |$1e^{-2}$|$4e^{-2}$|$3.1e^{-1}$|$2.4e^{0}$|$1.9e^{1}$| $1.5e^{2}$|
> > > |---|---|---|---|---|---|---|
> > > MM KL-UOT CPU|0.05|4.45|99.63|301.46|461.48|273.84|
> > > MM KL-UOT GPU|0.01|0.25|4.43|13.98|28.91|54.18|
> > > GPU speedup over CPU|9 x|18 x|22 x|22 x|16 x|5 x|
> > >
> > > The results show that running the algorithm is about 5 to 20 times faster than running on CPU.
> > > We propose to add those results on an updated version of the paper, together with the results associated to the $\ell_2$-UOT (for which we expect similar speed improvements).

---

> > > > ### Comment · Reviewer_42ef · 2021-08-29
> > > > **I raised the score**
> > > >
> > > > Thank you for your additional experimental results.  I think those are very good results.
> > > >
> > > > I raised the score from 5 to 6.

---

### Official Review · Reviewer_9N5u · 2021-07-17

**Rating:** 6
**Confidence:** 4

**Summary:**

The authors propose to reformulate the unbalanced Optimal Transport (e.g., Kullback-Leibler and $\ell_2$-regularization) and non-negative penalized linear regression with $\ell_1$-regularization. For $\ell_2$-regularized UOT, the authors derive the regularization path.


**Limitations And Societal Impact:**

Yes

**Main Review:**

It is interesting to reformulate the UOT as a non-negative penalized linear regression with $\ell_1$ regularization and leverage the tools in Lasso problem to derive the regularization path.

The majorization-minimization (MM) algorithmic approach for UOT seems elegant with the multiplicative update (naturally handing the non-negative constraint). However, there is no theoretical analysis to compare this MM algorithmic approach for the proposed reformulation with approaches in the literature (for KL/$\ell_2^2$ regularized UOT). Although the authors did the empirical comparison in Figure 3 (and recall that the complexity of L-BFGS is about $O(n^2)$), it seems unclear why the proposed MM algorithmic approach gives more advantages. It is better in case the authors elaborate with more details.

The regularization path for $\ell_2$-penalized UOT is interesting. However, it seems that the authors may need to discuss Algorithm 1 in more detail. For examples,
+ Can the stoping condition in the while-loop be obtained?
+ How to update the set A? (How to compute the set $A_{k+1}$?

For the right plot in Figure 1 (left), it may mislead the readers for the curve between $\lambda_7$ and $\infty$? (monotonically increasing?)
+ Do the authors compute the standard OT at $\infty$ or use Algorithm 1?

In line 290-292, the authors discuss the contribution with the SInkhorn-like multiplicative algorithm for UOT? However, it is unclear why the Sinkhorn-like algorithm is important and favoring? What is its advantage to compare with the L-BFGS (e.g., in Blondel et al.)

Overall, I think the submission is interesting. However, there are some points (as mentioned above) that may be needed further elaboration. Besides the regularization path results (for $\ell_2$ regularization), it is still unclear yet about the advantage of the MM algorithmic approach for the proposed reformulation with approaches in the literature.

---------After the rebuttal-----------

I thank the authors for the response. Overall, I think the proposed reformulated UOT to connect with the LASSO problem is interesting. I hope that the authors address those raised concerns, especially those about the MM algorithm in the updated version. So, I keep learning on the positive side.

**Time Spent Reviewing:**

8 hours

---

> ### Author Response · Authors · 2021-08-10
> **Answer to Reviewer 9N5u**
>
> ### No theoretical analysis of the MM algorithm
>
> The theoretical study of MM algorithms have been presented in the literature (see Dhillon and S. Sra (2005), Sun et al. (2017), Zhao and Tan (2018)). Under mild-conditions satisfied by our studied UOT, MM algorithms  are  guaranteed  to converge and since our porblms are convex to a global minima.
>
> We propose to add those references and a better explanatio in the updated version of the paper. Note that we propose the first algorithm that allows solving a KL-relaxed UOT problem with no additional entropic regularisation, which is of high interest when a sparse transport plan is sought.
>
>
> Sun, Ying, Prabhu Babu, and Daniel P. Palomar. "Majorization-minimization algorithms in signal processing, communications, and machine learning." IEEE Transactions on Signal Processing 65.3 (2016): 794-816
>
> R. Zhao and V. Y. F. Tan, "A Unified Convergence Analysis of the Multiplicative Update Algorithm for Regularized Nonnegative Matrix Factorization," in IEEE Transactions on Signal Processing, vol. 66, no. 1, pp. 129-138, 1 Jan.1, 2018, doi: 10.1109/TSP.2017.2757914.
>
> ### "However, it is unclear why the Sinkhorn-like algorithm is important and favoring? What is its advantage to compare with the L-BFGS (e.g., in Blondel et al.)"
>
> According to Fig 3, right panel, we have a very important gain with MM in the small $\lambda$ regime that can be of interest in practice (to detect out of distribution samples for instance). In this case LBFGS is slow because the problem is less and less quadratic/non linear and the quasi newton approximation makes less and less sense.
>
> We also think that the MM algorithm leads to very simple updates that can be implemented with just a few lines of code and use only matrix and pointwise products that have been heavily optimized on modern hardware (CPU and GPU) whereas more involved algorithms such as LBFGS will require more complex and hard to differentiate steps such as line search operations.
>
> ### ℓ2-penalized UOT and more discussion about Algorithm 1
> We provide here some additional details that we propose to add in an updated version of the paper.
>
> + 3.1 "Can the stoping condition in the while-loop be obtained?"
>
> This condition is testing the marginal constraints for the exact OT problem that will be true when $\lambda=\infty$. Note that at each iteration we have an increase of $\lambda_k$ (with a combinatorial but finite number of  iterations in the worst case and well known in LARS). So the stopping condition will be obtained in practice (up to numerical precision of course). In practice we reach $\lambda=\infty$ when all the KKT conditions are respected with $\mathbf{t}_{\mathcal{A}}=\widetilde{\mathbf{m}}_{\mathcal{A}}$ in Equation (9).
>
> + 3.2 "How to update the set A? (How to compute the set $A_{k+1}$?"
>
> As discussed maybe a bit shortly in the paper in section 3.2 the update of $A_{k+1}$ depends on the KKT conditions, either a component will leave $A$ ($\lambda_r$ equation (10)) or a new component will become active ($\lambda_a$ equation (11)). The selection between those two moves is done by finding which of the recoverd $\lambda_r$ and $\lambda_a$ is the smallest and the component that will be removed (resp. added) is the one corresponding to the argmin in equation (10) (resp. (11)). This part might not be detailled enough (especially around line 204) due to lack of space. We will clarify this in the final version.
>
> More information about the implementation is provided in the file code/solvers/solvers_L2_UOT.py in the supplementary. It shows the computation  of the two lambdas are computed in line 152 and the removing/adding of the components is done in lines 183-192.
>
> + 3.3 "Figure 1 (left), it may mislead the readers for the curve between $\lambda_7$ and $\infty$"
>
> We agree that the figure might be misleading but we wanted to also show the values for the exact OT that is recovered for $\lambda=\infty$. Note that the variation between two $\lambda$ values is linear (in $1/\lambda$) so yes the increasing is monotonic on the last segment $[\lambda_7,\infty]$. We  will make a more detailled caption in the final version.
>
> + 3.4 "Do the authors compute the standard OT or use Algorithm 1?"
>
> The Figure is generated using the file Figure 1.ipynb in supplementary where we indeed use the Exact OT solution returned by Algorithm 1. As shown in the paper, Algorithm 1 returns the same solution at the last iteration as an exact OT solver (from POT toolbox) up to numerical precision.

---

### Official Review · Reviewer_1XGy · 2021-07-24

**Rating:** 6
**Confidence:** 3

**Summary:**

Authors in this paper flattened the matrix-form unbalanced OT and reformulated it as a linear regression regularized by either the KL divergence or $\ell_2$ between the transport plan and two marginals. Then the problem converts to an MM problem and the authors derived the optimization steps for solving them, especially the one with $\ell_2$ in which case the authors proposed an algorithm to regress the optimal combinations of the transport plan and the regularization strength. Numerical results show that the new formulation leads to faster computation.

**Limitations And Societal Impact:**

Yes.

**Main Review:**

+
The proposed method is novel and all the claims in the paper seem well supported by theoretical and empirical results.

-
Reformulating $\ell_2$-regularized UOT to a Lasso regression inevitably promotes sparsity. Adding the transportation cost $c^Tt$, the resulting transporting map $t^k$ concentrates on the nearest samples of the two domains like what $Figure\ 2$ shows, especially when $\lambda$ is small. When $\lambda_1 = \lambda_2$, it becomes more apparent that $Algorithm\ 1$ is selecting a subset of the samples and finding a balanced transport. Although it still aligns with the original formulation of unbalanced OT (Benamou, 2003), but my understanding of the practicality of unbalanced OT is that it still transports all the mass but allows the mass to grow or shrink during transport. Now, the proposed method seems more like to transport a subset of mass controlled by $\lambda$.

I am also reluctant to accept the wording that $\lambda \rightarrow \infty$ recovers a balanced OT because balanced OT requires the mass on both sides to equal, i.e. $ \mathbb{1}a =  \mathbb{1}b$ and such a T does not exist when $\lambda \rightarrow \infty$ and $ \mathbb{1}a \neq  \mathbb{1}b$. In this sense, the experiment in Figure 2 feels cheesy to me because the two marginals are balanced, and $Algorithm\ 1$ feels more like gradually finding a balanced OT by including more samples and excluding fewer outliers from the two domains.

The motivation behind computing the "regularization path" is not clear in the beginning of the paper. Only when I saw the application to unbalanced DA did I get to know its usage but I'm still not sure what motivated the authors to get such as path. It seems the paper has a very similar motivation to those of Mukherjee et al. (line 416) and Balaji et al.

Balaji, Yogesh and Chellappa, Rama and Feizi, Soheil, Robust Optimal Transport with Applications in Generative Modeling and Domain Adaptation, Advances in Neural Information Processing Systems, 2020


**Time Spent Reviewing:**

3

---

> ### Author Response · Authors · 2021-08-10
> **Answer to Reviewer 1XGy**
>
> ### The proposed method seems to find a balanced OT transport on a subset of samples
>
>
> When the support is known (that allows finding the samples that belong to the solution), we show in the paper how to compute the associated transport plan. But no, it *does not* correspond to a balanced OT between the samples of the support. As an illustration, all the samples have a non-zeros marginal for $\lambda $ values greater than 117.18 (only for the first run, we have actually 5 runs in Figure 3) in Figure 3 but the transport plan still evolves when $\lambda $ continues to grow.
>
>
> ### “Unbalanced OT is that it still transport all the mass”
>
>
> We don’t quite understand the meaning of this statement. UOT only transports a portion of mass, which is controlled by $\lambda$. When relaxing the constraints on the marginals, as UOT is a minimization problem, it leads to more sparse marginals in the solution (not adding some mass that would increase the transportation cost).
>
> ### Regularization path recovers the balanced OT when $\lambda \rightarrow \infty$
>
>
> UOT comes down to standard OT only when the mass of the source and target distributions are equal. We will make more clear in the final version that we assume in thIs statement that $\sum_i a_j = \sum_j b_j$. If this assumption is not satisfied, we agree that one cannot recover the balanced OT solution as the solution does not exist.
>
> ### Motivation of regularization path
>
>
> Having a continuity of solutions for the whole regularization path is a tremendous result in machine learning (ML) as the hyper-parameters of all ML methods have to be validated in practice which require a lot of computation power. When looking at the center panel of Figure 3 we can see that solving the UOT for large $\lambda$s is slightly faster with Lasso solvers (which is also a contribution since no research work formulates the UOT with quadratic divergence  as a Lasso), but those solvers only provide  a solution for a unique value of $\lambda$. Estimating the relevant $\lambda$ given a UOT task might be very costly whereas the regularization path provides the whole solution path for a fairly limited numerical cost. Finally, searching for "regularization path" in Google scholar returns more than 4000 papers which illustrates the interest of the approach in practice and our paper is the first to provide  a regularization path algorithm to solve OT problem to the best of our knowledge.
>
> Regarding the similarity with Mukherjee et al., they focus on the special case of TV unbalanced while we study the L2 and KL unbalanced OT. Balaji et al. focus on Robust OT and use neural network approximation to solve the problem whereas we propose solvers for exact resolution. Furthermore, it allows one to easily cross validate the $\lambda$ parameter, which is an important Issue and eases the optimal parameter search (as in Mukherjee et al. for instance, in which they state that It can be hard to pick a good $\lambda$).

---

> > ### Comment · Reviewer_1XGy · 2021-08-30
> > **Concerns addressed**
> >
> > After reading literatures about regularization path, I find the paper is convincing. The authors addressed my concerns. I have also read other reviewers' comments and the response. I am keeping a positive rating.

---

### Author Response · Authors · 2021-08-10
**Answer to all reviewers**

We thank the reviewers for their insightful comments that will help us improving the paper, especially by pointing out some points that may need further elaboration. We were constraint by space and made some choices; we believe that an updated version will allow providing further details on the method.
We are glad that the reviewers found the submission interesting (Rev 9N5u, 42ef), original and well written (Rev 42ef). They pointed out that the proposed method is novel, well supported theoretically and empirically (Rev 1XGy) and efficient (7tpv).

---

> ### Author Response · Authors · 2021-08-26
> **Further discussion**
>
> Dear reviewers,
>
> we hope that we have fully understood and addressed your concerns; if not or if new questions come up, we are willing to answer any comments.
>
> Best,

---

### Decision · Program_Chairs · 2021-09-27

**Decision:**

Accept (Poster)

**Comment:**

The authors consider the unbalanced optimal transport (OT) problem and its semi-relaxed variants. They then consider various optimization approaches that come from the mirror descent family by choosing an appropriate Bregman divergence. In particular, they consider the entropic and the ell2 mirror maps.

In the ell2 case, the authors connect the optimization problem to the LARS optimization, which has been quite successful in the sparse recovery literature. They propose a regularization path or homotopy algorithm for the unbalanced OT plans and demonstrate its performance numerically.

The developments in the paper mostly rely on recognizing the LASSO structure in the unbalanced OT problem and then applying existing techniques to it. In the worst case, this approach has exponential complexity but the authors argue that the [Mairal and Yu 2012] approach can be applied to obtain eps-approximate primal dual gap as an afterthought. Unfortunately, there is also no follow up to back up how this epsilon guarantee to the quality of the OT map.